# REVISIT AND OUTSTRIP ENTITY ALIGNMENT: A PERSPECTIVE OF GENERATIVE MODELS

**Lingbing Guo**[1,2,3][*] **Zhuo Chen**[1,2,3][*] **Jiaoyan Chen**[4]**, Yin Fang**[1,2,3]**, Wen Zhang**[5,2,3][†]

**Huajun Chen**[1,2,3][†]

[1]College of Computer Science and Technology, Zhejiang University
[2]Zhejiang University - Ant Group Joint Laboratory of Knowledge Graph
[3]Alibaba-Zhejiang University Joint Reseach Institute of Frontier Technologies
[4]Department of Computer Science, The University of Manchester
[5]School of Software Technology, Zhejiang University

## ABSTRACT

Recent embedding-based methods have achieved great successes in exploiting entity alignment from knowledge graph (KG) embeddings of multiple modalities. In this paper, we study embedding-based entity alignment (EEA) from a perspective of generative models. We show that EEA shares similarities with typical generative models and prove the effectiveness of the recently developed generative adversarial network (GAN)-based EEA methods theoretically. We then reveal that their incomplete objective limits the capacity on both entity alignment and entity synthesis (i.e., generating new entities). We mitigate this problem by introducing a generative EEA (GEEA) framework with the proposed mutual variational autoencoder (M-VAE) as the generative model. M-VAE enables entity conversion between KGs and generation of new entities from random noise vectors. We demonstrate the power of GEEA with theoretical analysis and empirical experiments on both entity alignment and entity synthesis tasks. The source code and datasets are available at github.com/zjukg/GEEA.

## 1 INTRODUCTION

As one of the most prevalent tasks in the knowledge graph (KG) area, entity alignment (EA) has recently made great progress and developments with the support of the embedding techniques (Chen et al., 2017; Sun et al., 2017; Zhang et al., 2019; Chen et al., 2020; Liu et al., 2021; Chen et al., 2022a;b; Guo et al., 2022a;b; Chen et al., 2024). By encoding the relational and other information into low-dimensional vectors, the embedding-based entity alignment (EEA) methods are friendly for development and deployment, and have achieved state-of-the-art performance on many benchmarks.

The objective of EA is to maximize the conditional probability $p(y|x)$, where $x$, $y$ are a pair of aligned entities belonging to source KG $\mathcal{X}$ and target KG $\mathcal{Y}$, respectively. If we view $x$ as the input and $y$ as the label (and vice versa), the problem can be solved by a discriminative model. To this end, we need an EEA model which comprises an encoder module and a fusion layer (Zhang et al., 2019; Chen et al., 2020; Liu et al., 2021; Chen et al., 2022a;b; Lin et al., 2022) (see Figure 1). The encoder module uses different encoders to encode multi-modal information into low-dimensional embeddings. The fusion layer then combines these sub-embeddings to a joint embedding as the output.

We also need a predictor, as shown in the yellow area in Figure 1. The predictor is usually independent of the EEA model and parameterized with neural layers (Chen et al., 2017; Guo et al., 2020) or based on the embedding distance (Sun et al., 2017; 2018). In either case, it learns the probability $p(y|x)$ where $p(y|x) = 1$ if the two entities $x$, $y$ are aligned and 0 otherwise. The difference lies primarily in data augmentation. The existing methods employ different strategies to construct more training data,

---

[*]Equal contribution
[†]Correspondence to: {zhang.wen, huajunsir}@zju.edu.cn

Figure 1: Illustration of embedding-based entity alignment. The modules in the blue area belong to the EEA model, while those in the yellow area belong to the predictor.

e.g., negative sampling (Chen et al., 2017; Sun et al., 2017; Wang et al., 2018) and bootstrapping (Sun et al., 2018; Pei et al., 2019a; Guo et al., 2022a).

In fact, entity alignment is not the ultimate aim of many applications. The results of entity alignment are used to enrich each other's KGs, but there are often entities in the source KG that do not have aligned counterparts in the target KG, known as *dangling entities* (Sun et al., 2021; Luo & Yu, 2022). For instance, a source entity *Star Wars (film)* may not have a counterpart in the target KG, which means we cannot directly enrich the target KG with the information of *Star Wars (film)* via entity alignment. However, if we can convert entities like *Star Wars (film)* from the source KG to the target KG, it would save a major expenditure of time and effort for many knowledge engineering tasks, such as knowledge integration and fact checking. Hence, we propose *conditional entity synthesis* to generate new entities for the target KG with the entities in the source KG as input. Additionally, generating new entities from random variables may contribute to the fields like Metaverse and video games where the design of virtual characters still relies on hand-crafted features and randomized algorithms (Khalifa et al., 2017; Lee et al., 2021). For example, modern video games feature a large number of non-player characters (NPCs) with unique backgrounds and relationships, which are essential for creating immersive virtual worlds. Designing NPCs is a laborious and complex process, and using the randomized algorithms often yields unrealistic results. By storing the information and relationships of NPCs in a KG, one can leverage even a small initial training KG to generate high-quality NPCs with coherent backgrounds and relationships. Therefore, we propose *unconditional entity synthesis* for generating new entities with random noise vectors as input.

We propose a generative EEA (abbr., GEEA) framework with the mutual variational autoencoder (M-VAE) to encode/decode entities between source and target KGs. GEEA is capable of generating concrete features, such as the exact neighborhood or attribute information of a new entity, rather than only the inexplicable embeddings as previous works have done (Pei et al., 2019a;b; Guo et al., 2022b). We introduce the prior reconstruction and post reconstruction to control the generation process. Briefly, the prior reconstruction is used to generate specific features for each modality, while the post reconstruction ensures these different kinds of features belong to the same entity. We conduct experiments to validate the performance of GEEA, where it achieves state-of-the-art performance in entity alignment and generates high-quality new entities in entity synthesis.

## 2 Revisit Embedding-based Entity Alignment

In this section, we revisit embedding-based entity alignment by a theoretical analysis of how the generative models contribute to entity alignment learning, and then discuss their limitations.

### 2.1 Preliminaries

**Entity Alignment** Entity alignment aims to find the implicitly aligned entity pairs $\{(x, y) | x \in \mathcal{X}, y \in \mathcal{Y}\}$, where $\mathcal{X}$, $\mathcal{Y}$ denote the source and target entity sets, and $(x, y)$ represents a pair of aligned entities referring to the same real-world object. An EEA model $\mathcal{M}$ uses a small number of aligned entity pairs $\mathcal{S}$ (a.k.a., seed alignment set) as training data to infer the remaining alignment pairs $\mathcal{T}$ in the testing set. We consider three different modalities: relational graphs $\mathcal{G}_x, \mathcal{G}_y$, attributes $\mathcal{A}_x, \mathcal{A}_y$, and images $\mathcal{I}_x, \mathcal{I}_y$. Other types of information can be also given as features for $\mathcal{X}$ and $\mathcal{Y}$.

For instance, the relational graph feature of an entity *Star Wars (film)* is represented as triplets, such as *(Star Wars (film), founded by, George Lucas)*. Similarly, the attribute feature is represented as

attribute triplets, e.g., *(Star Wars (film), title, Star Wars (English))*. For the image feature, we follow the existing multi-modal EEA works to use a constant pretrained embedding from a vision model as the image feature of *Star Wars (film)* (Liu et al., 2021; Lin et al., 2022). The EEA model $\mathcal{M}$ takes the above multi-modal features $x = (g_x, a_x, i_x, ...)$ as input, where $g_x, a_x, i_x$ denote the relational graph information, attribute information and image information of $x$, respectively. The output consists of the embeddings for each modality (i.e., sub-embeddings) and a final output embedding $\mathbf{x}$ (i.e., joint embedding) that combines all modalities:

$$\mathbf{x} = \mathcal{M}(x) = Linear(Concat(\mathcal{M}_g(g_x), \mathcal{M}_a(a_x), \mathcal{M}_i(i_x), ...)) \tag{1}$$

$$= Linear(Concat(\mathbf{g}_x, \mathbf{a}_x, \mathbf{i}_x, ...)), \tag{2}$$

where $\mathcal{M}_g$, $\mathcal{M}_a$, and $\mathcal{M}_i$ denote the EEA encoders for different modalities (also see Figure 1). $\mathbf{g}_x$, $\mathbf{a}_x$, and $\mathbf{i}_x$ denote the embeddings of different modalities. Similarly, we obtain $\mathbf{y}$ by $\mathbf{y} = \mathcal{M}(y)$.

**Entity Synthesis**  We consider two entity synthesis tasks: conditional entity synthesis and unconditional entity synthesis. Conditional entity synthesis aims to generate entities in the target KG with the dangling entities in the source KG as input. Formally, the model takes an entity $x$ as input and convert it to an entity $y_{x \to y}$ for the target KG. It should also produce the corresponding concrete features, such as neighborhood and attribute information specific to the target KG. On the other hand, the unconditional entity synthesis involves generating new entities in the target KG with random noise variables as input. Formally, the model takes a random noise vector $\mathbf{z}$ as input and generate a target entity embedding $\mathbf{y}_{z \to y}$ which is then converted back to concrete features.

For instance, to reconstruct the neighborhood (or attribute) information of *Star Wars (film)* from its embedding, we can leverage a decoder module to convert the embedding into a probability distribution of all candidate entities (or attributes). As the image features are constant pretrained embeddings, we can use the image corresponding to the nearest neighbor of the reconstructed image embedding of *Star Wars (film)* as the output image.

**Generative Models**  Generative models learn the underlying probability distribution $p(x)$ of the input data $x$. Take variational autoencoder (VAE) (Kingma & Welling, 2013) as an example, the encoding and decoding processes can be defined as:

$$\mathbf{h} = \text{Encoder}(\mathbf{x}) \qquad \text{(Encoding)} \tag{3}$$

$$\mathbf{z} = \mu + \sigma \odot \epsilon = \text{Linear}_\mu(\mathbf{h}) + \text{Linear}_\sigma(\mathbf{h}) \odot \epsilon \qquad \text{(Reparameterization Trick)} \tag{4}$$

$$\mathbf{x}_{x \to x} = \text{Decoder}(\mathbf{z}) \qquad \text{(Decoding)}, \tag{5}$$

where $\mathbf{h}$ is the hidden output. VAE uses the reparameterization trick to rewrite $\mathbf{h}$ as coefficients $\mu$, $\sigma$ in a deterministic function of a noise variable $\epsilon \in \mathcal{N}(\epsilon; \mathbf{0}, \mathbf{I})$, to enable back-propagation. $\mathbf{x}_{x \to x}$ denotes that this reconstructed entity embedding is with $x$ as input and for $x$. VAE generates new entities by sampling a noise vector $\mathbf{z}$ and converting it to $\mathbf{x}$.

## 2.2 EEA Benefits from the Generative Objectives

Let $x \sim \mathcal{X}, y \sim \mathcal{Y}$ be two entities sampled from the entity sets $\mathcal{X}, \mathcal{Y}$, respectively. The main target of EEA is to learn a predictor that estimates the conditional probability $p_\theta(\mathbf{x}|\mathbf{y})$ (and reversely $p_\theta(\mathbf{y}|\mathbf{x})$), where $\theta$ represents the parameter set. For simplicity, we assume that the reverse function $p_\theta(\mathbf{y}|\mathbf{x})$ shares the same parameter set with $p_\theta(\mathbf{x}|\mathbf{y})$.

Now, suppose that one wants to learn a generative model for generating entity embeddings:

$$\log p(\mathbf{x}) = \log p(\mathbf{x}) \int p_\theta(\mathbf{y}|\mathbf{x}) d\mathbf{y} \tag{6}$$

$$= \mathbb{E}_{p_\theta(\mathbf{y}|\mathbf{x})} \left[ \log \frac{p(\mathbf{x}, \mathbf{y})}{p_\theta(\mathbf{y}|\mathbf{x})} \right] + D_{\text{KL}}(p_\theta(\mathbf{y}|\mathbf{x}) \parallel p(\mathbf{y}|\mathbf{x})), \tag{7}$$

where the left-hand side of Equation (7) is the evidence lower bound (ELBO) (Kingma & Welling, 2013), and the right-hand side is the Kullback-Leibler (KL) divergence (Kullback & Leibler, 1951) between our parameterized distribution $p_\theta(\mathbf{y}|\mathbf{x})$ (i.e., the predictor) and the true distribution $p(\mathbf{y}|\mathbf{x})$.

In typical generative learning, $p(\mathbf{y}|\mathbf{x})$ is intractable because $\mathbf{y}$ is a noise variable sampled from a normal distribution, and thus $p(\mathbf{y}|\mathbf{x})$ is unknown. However, in EEA, we can obtain a few samples by

using the training set, which leads to a classical negative sampling loss (Sun et al., 2017; Cao et al., 2019; Zhang et al., 2019; Chen et al., 2020; Guo et al., 2020; Sun et al., 2020a; Liu et al., 2021; Chen et al., 2022a;b; Guo et al., 2022a;b; Lin et al., 2022):

$$\mathcal{L}_{\mathrm{ns}} = \sum_i [-\log(p_\theta(\mathbf{y}^i|\mathbf{x}^i)p(\mathbf{y}^i|\mathbf{x}^i)) + \frac{1}{N_{\mathrm{ns}}} \sum_{j \neq i} \log\left(p_\theta(\mathbf{y}^j|\mathbf{x}^i)(1 - p(\mathbf{y}^j|\mathbf{x}^i))\right)], \tag{8}$$

where $(\mathbf{y}^i, \mathbf{x}^i)$ denotes a pair of aligned entities in the training data. The randomly sampled entity $\mathbf{y}^j$ is regarded as the negative entity. $i, j$ are the entity IDs. $N_{\mathrm{ns}}$ is the normalization constant. Here, $\mathcal{L}_{\mathrm{ns}}$ is formulated as a cross-entropy loss with the label $p(\mathbf{y}^j|\mathbf{x}^i)$ defined as:

$$p(\mathbf{y}^j|\mathbf{x}^i) = \begin{cases} 0, & \text{if } i \neq j, \\ 1, & \text{otherwise} \end{cases} \tag{9}$$

Given that EEA typically uses only a small number of aligned entity pairs for training, the observation of $p(\mathbf{y}|\mathbf{x})$ may be subject to bias and limitations. To alleviate this problem, the recent GAN-based methods (Pei et al., 2019a;b; Guo et al., 2022b) propose leveraging entities outside the training set for unsupervised learning. The common idea behind these methods is to make the entity embeddings from different KGs indiscriminative to a discriminator, such that the underlying aligned entities shall be encoded in the same way and have similar embeddings. To formally prove this idea, we dissect the ELBO in Equation (7) as follows:

$$\mathbb{E}_{p_\theta(\mathbf{y}|\mathbf{x})}\left[\log\frac{p(\mathbf{x}, \mathbf{y})}{p_\theta(\mathbf{y}|\mathbf{x})}\right] = \mathbb{E}_{p_\theta(\mathbf{y}|\mathbf{x})}\left[\log p_\theta(\mathbf{x}|\mathbf{y})\right] - D_{\mathrm{KL}}(p_\theta(\mathbf{y}|\mathbf{x}) \parallel p(\mathbf{y})) \tag{10}$$

The complete derivation in this section can be found in Appendix A.1. Therefore, we have:

$$\log p(\mathbf{x}) = \underbrace{\mathbb{E}_{p_\theta(\mathbf{y}|\mathbf{x})}\left[\log p_\theta(\mathbf{x}|\mathbf{y})\right]}_{\text{reconstruction term}} - \underbrace{D_{\mathrm{KL}}(p_\theta(\mathbf{y}|\mathbf{x}) \parallel p(\mathbf{y}))}_{\text{distribution matching term}} + \underbrace{D_{\mathrm{KL}}(p_\theta(\mathbf{y}|\mathbf{x}) \parallel p(\mathbf{y}|\mathbf{x}))}_{\text{prediction matching term}} \tag{11}$$

The first term aims to reconstruct the original embedding $\mathbf{x}$ based on $\mathbf{y}$ generated from $\mathbf{x}$, which has not been studied in existing discriminative EEA methods (Guo et al., 2020; Liu et al., 2021; Lin et al., 2022). The second term enforces the distribution of $\mathbf{y}$ conditioned on $\mathbf{x}$ to match the prior distribution of $\mathbf{y}$, which has been investigated by the GAN-based EEA methods (Pei et al., 2019a;b; Guo et al., 2022b). The third term represents the main objective of EEA (as described in Equation (8) where the target $p(\mathbf{y}|\mathbf{x})$ is partially observed).

Note that, $p(\mathbf{x})$ is irrelevant to our parameter set $\theta$ and can be treated as a constant during optimization. Consequently, maximizing the ELBO (i.e., maximizing the first term and minimizing the second term) will result in minimizing the third term:

**Proposition 1.** *Maximizing the reconstruction term and/or minimizing the distribution matching term subsequently minimizes the EEA prediction matching term.*

The primary objective of EEA is to minimize the prediction matching term. Proposition 1 provides theoretical evidence that the generative objectives naturally contribute to the minimization of the EEA objective, thereby enhancing overall performance.

## 2.3 THE LIMITATIONS OF GAN-BASED EEA METHODS

The GAN-based EEA methods leverage a discriminator to discriminate the entities from one KG against those from another KG. Supposed that $\mathbf{x}, \mathbf{y}$ are embeddings produced by an EEA model $\mathcal{M}$, sampled from the source KG and the target KG, respectively. The GAN-based methods train a discriminator $\mathcal{D}$ to distinguish $\mathbf{x}$ from $\mathbf{y}$ (and vice versa), with the following objective:

$$\underset{\mathbf{x}, \mathbf{y}, \psi}{\operatorname{argmax}}\left[\mathbb{E}_{x \sim \mathcal{X}} \log \mathcal{D}_\phi(\mathcal{M}_\psi(x)) + \mathbb{E}_{y \sim \mathcal{Y}} \log \mathcal{D}_\phi(\mathcal{M}_\psi(y))\right] \quad \text{(Generator)} \tag{12}$$

$$+ \underset{\phi}{\operatorname{argmax}}\left[\mathbb{E}_{x \sim \mathcal{X}} \log \mathcal{D}_\phi(\mathcal{M}_\psi(x)) + \mathbb{E}_{y \sim \mathcal{Y}} \log(1 - \mathcal{D}_\phi(\mathcal{M}_\psi(y)))\right] \quad \text{(Discriminator)} \tag{13}$$

Here, the EEA model $\mathcal{M}$ takes entities $x, y$ as input and produces the output embeddings $\mathbf{x}, \mathbf{y}$, respectively. $\mathcal{D}$ is the discriminator that learns to predict whether the input variable is from the target distribution. $\phi, \psi$ are the parameter sets of $\mathcal{M}, \mathcal{D}$, respectively.

It is important to note that both $\mathbf{x} = \mathcal{M}_\psi(x)$ and $\mathbf{y} = \mathcal{M}_\psi(y)$ do not follow a fixed distribution (e.g., a normal distribution). They are learnable vectors during training, which is significantly different from the objective of a typical GAN, where variables like $\mathbf{x}$ (e.g., an image) and $\mathbf{z}$ (e.g., sampled from a normal distribution) have deterministic distributions. Consequently, the generator in Equation (12) can be overly strong, allowing $\mathbf{x}, \mathbf{y}$ to be consistently mapped to plausible positions to deceive $\mathcal{D}$.

Therefore, one major issue with the existing GAN-based methods is *mode collapse* (Srivastava et al., 2017; Pei et al., 2019b; Guo et al., 2022b). Mode collapse often occurs when the generator (i.e., the EEA model in our case) over-optimizes for the discriminator. The generator may find some outputs appear most plausible to the discriminator and consistently produces those outputs. This is harmful for EEA as irrelevant entities are encouraged to have similar embeddings. We argue that *mode collapse* is more likely to occur in the existing GAN-based EEA methods, which is why they often use a very small weight (e.g., 0.001 or less) to optimize the generator against the discriminator (Pei et al., 2019b; Guo et al., 2022b).

Another limitation of the existing GAN-based methods is their inability to generate new entities. The generated target entity embedding $\mathbf{y}_{x \to y}$ cannot be converted back to the native concrete features, such as the neighborhood $\{y^1, y^2, ...\}$ or attributes $\{a^1, a^2, ...\}$.

## 3 GENERATIVE EMBEDDING-BASED ENTITY ALIGNMENT

### 3.1 MUTUAL VARIATIONAL AUTOENCODER

In many generative tasks, such as image synthesis, the conditional variable (e.g., a textual description) and the input variable (e.g., an image) differ in modality. However, in our case, they are entities from different KGs. Therefore, we propose mutual variational autoencoder (M-VAE) for efficient generation of new entities. One of the most important characteristics of M-VAE lies in the variety of the encode-decode process. It has four different flows:

The first two flows are used for self-supervised learning, i.e., reconstructing the input variables:

$$\mathbf{x}_{x \to x}, \mathbf{z}_{x \to x} = VAE(\mathbf{x}), \quad \mathbf{y}_{y \to y}, \mathbf{z}_{y \to y} = VAE(\mathbf{y}), \quad \forall x, \forall y, x \in \mathcal{X}, y \in \mathcal{Y} \qquad (14)$$

We use the subscript $_{x \to x}$ to denote the flow is from $x$ to $x$, and similarly for $_{y \to y}$. $\mathbf{z}_{x \to x}, \mathbf{z}_{y \to y}$ are the latent variables (as defined in Equation 4) of the two flows, respectively. In EEA, the majority of alignment pairs are unknown, but all information of the entities is known. Thus, these two flows provide abundant examples to train GEEA in a self-supervised fashion.

The latter two flows are used for supervised learning, i.e., reconstructing the mutual target variables:

$$\mathbf{y}_{x \to y}, \mathbf{z}_{x \to y} = VAE(\mathbf{x}), \quad \mathbf{x}_{y \to x}, \mathbf{z}_{y \to x} = VAE(\mathbf{y}), \quad \forall (x, y) \in \mathcal{S}. \qquad (15)$$

It is worth noting that we always share the parameters of VAEs across all flows. We wish the rich experience gained from reconstructing the input variables (Equation (14)) can be flexibly conveyed to reconstructing the mutual target (Equation (15)).

### 3.2 DISTRIBUTION MATCH

The existing GAN-based methods directly minimize the KL divergence (Kullback & Leibler, 1951) between two embedding distributions, resulting in the over-optimization of generator and incapability of generating new entities. In this paper, we propose to draw support from the latent noise variable $\mathbf{z}$ to avoid these two issues. The distribution match loss is defined as follows:

$$\mathcal{L}_{\text{kld}} = D_{\text{KL}}(p(\mathbf{z}_{x \to x}), p(\mathbf{z}^*)) + D_{\text{KL}}(p(\mathbf{z}_{y \to y}), p(\mathbf{z}^*)). \qquad (16)$$

where $p(\mathbf{z}_{x \to x})$ denotes the distribution of $\mathbf{z}_{x \to x}$, and $p(\mathbf{z}^*)$ denotes the target normal distribution. We do not optimize the distributions of $\mathbf{z}_{x \to y}, \mathbf{z}_{y \to x}$ in the latter two flows, because they are sampled from seed alignment set $\mathcal{S}$, a (likely) biased and small training set.

Minimizing $\mathcal{L}_{\text{kld}}$ can be regarded as aligning the entity embeddings from respective KGs to a fixed normal distribution. We provide a formal proof that the entity embedding distributions of two KGs will be aligned although we do not implicitly minimize $D_{\text{KL}}(p(\mathbf{x}), p(\mathbf{y}))$:

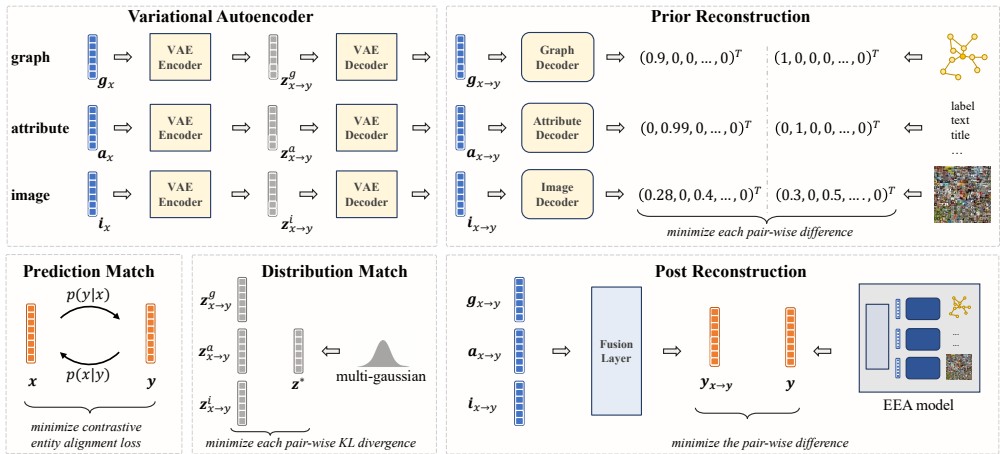

Figure 2: The workflow of GEEA. Top: different sub-VAEs process different sub-embeddings, and the respective decoders convert the sub-embeddings back to concrete features. Bottom-left: the entity alignment prediction loss is retained. Bottom-center: the latent variables of sub-VAEs are used for distribution matching. Bottom-right: The reconstructed sub-embeddings are feed into the fusion layer in the EEA model to produce the reconstructed joint embedding for post reconstruction.

**Proposition 2.** *Let $\mathbf{z}^*$, $\mathbf{z}_{x \to x}$, $\mathbf{z}_{y \to y}$ be the normal distribution, and the latent variable distributions w.r.t. $\mathcal{X}$ and $\mathcal{Y}$, respectively. Jointly minimizing the KL divergence $D_{\mathrm{KL}}(p(\mathbf{z}_{x \to x}), p(\mathbf{z}^*))$, $D_{\mathrm{KL}}(p(\mathbf{z}_{y \to y}), p(\mathbf{z}^*))$ will contribute to minimizing $D_{\mathrm{KL}}(p(\mathbf{x}), p(\mathbf{y}))$:*

$$D_{\mathrm{KL}}(p(\mathbf{x}), p(\mathbf{y})) \propto D_{\mathrm{KL}}(p(\mathbf{z}_{x \to x}), p(\mathbf{z}^*)) + D_{\mathrm{KL}}(p(\mathbf{z}_{y \to y}), p(\mathbf{z}^*)) \tag{17}$$

*Proof.* Please see Appendix A.2. □

### 3.3 PRIOR RECONSTRUCTION

The prior reconstruction aims to reconstruct the sub-embedding of each modality and recover the original concrete feature from the sub-embedding. Take the relational graph information of flow $x \to y$ as an example, we first employ a sub-*VAE* to process the input sub-embedding:

$$\mathbf{g}_{x \to y}, \mathbf{z}_{x \to y}^g = VAE_g(\mathbf{g}_x) \tag{18}$$

where $VAE_g$ denotes the variational autoencoder for relational graph information. $\mathbf{g}_x$ is the graph embedding of $x$, and $\mathbf{g}_{x \to y}$ is the reconstructed graph embedding for $y$ based on $x$. $\mathbf{z}_{x \to y}^g$ is the corresponding latent variable. To recover the original features (i.e., the neighborhood information of $y$), we consider a prediction loss defined as:

$$\mathcal{L}_{\mathbf{g}_{x \to y}} = g_y \log Decoder_g(\mathbf{g}_{x \to y}) + (1 - g_y) \log(1 - Decoder_g(\mathbf{g}_{x \to y})) \tag{19}$$

Here, $\mathcal{L}_{\mathbf{g}_{x \to y}}$ is a binary cross-entropy (BCE) loss. We employ a decoder $Decoder_g$ to convert the reconstructed sub-embedding $\mathbf{g}_{x \to y}$ to a probability estimation regarding the neighborhood of $y$.

### 3.4 POST RECONSTRUCTION

We propose post reconstruction to ensure the reconstructed features of different modalities belong to the same entity. We re-input the reconstructed sub-embeddings $\{\mathbf{g}_{x \to y}, \mathbf{a}_{x \to y}, ...\}$ to the fusion layer (defined in the EEA model $\mathcal{M}$) to obtain a reconstructed joint embedding $\mathbf{y}_{x \to y}$. We then employs mean square error (MSE) loss to match the reconstructed joint embedding with the original one:

$$\mathbf{y}_{x \to y} = Fusion(\{\mathbf{g}_{x \to y}, \mathbf{a}_{x \to y}, ...\}), \quad \forall (x, y) \in \mathcal{S} \tag{20}$$

$$\mathcal{L}_{x \to y} = MSE(\mathbf{y}_{x \to y}, NoGradient(\mathbf{y})), \quad \forall (x, y) \in \mathcal{S}, \tag{21}$$

where $\mathcal{L}_{x \to y}$ denotes the post reconstruction loss for the reconstructed joint embedding $\mathbf{y}_{x \to y}$. *Fusion* represents the fusion layer in $\mathcal{M}$, and *MSE* is the mean square error. We use the copy value of the original joint embedding $NoGradient(\mathbf{y})$ to avoid $\mathbf{y}$ inversely match $\mathbf{y}_{x \to y}$.

Table 1: Entity alignment results on DBP15K datasets, without surface information and iterative strategy. ↑: higher is better; ↓: lower is better. Average of 5 runs, the same below.

| Models | DBP15K$_{\text{ZH-EN}}$ | | | DBP15K$_{\text{JA-EN}}$ | | | DBP15K$_{\text{FR-EN}}$ | | |
|---|---|---|---|---|---|---|---|---|---|
| | Hits@1↑ | Hits@10↑ | MRR↑ | Hits@1↑ | Hits@10↑ | MRR↑ | Hits@1↑ | Hits@10↑ | MRR↑ |
| MUGNN (Cao et al., 2019) | .494 | .844 | .611 | .501 | .857 | .621 | .495 | .870 | .621 |
| AliNet (Sun et al., 2020a) | .539 | .826 | .628 | .549 | .831 | .645 | .552 | .852 | .657 |
| decentRL (Guo et al., 2020) | .589 | .819 | .672 | .596 | .819 | .678 | .602 | .842 | .689 |
| EVA (Liu et al., 2021) | .680 | .910 | .762 | .673 | .908 | .757 | .683 | .923 | .767 |
| MSNEA (Chen et al., 2022a) | .601 | .830 | .684 | .535 | .775 | .617 | .543 | .801 | .630 |
| MCLEA (Lin et al., 2022) | .715 | .923 | .788 | .715 | .909 | .785 | .711 | .909 | .782 |
| NeoEA (MCLEA) (Guo et al., 2022b) | .723 | .924 | .796 | .721 | .909 | .789 | .717 | .910 | .787 |
| GEEA | **.761** | **.946** | **.827** | **.755** | **.953** | **.827** | **.776** | **.962** | **.844** |

Table 2: Results on FB15K-DB15K and FB15K-YAGO15K datasets.

| Models | # Paras (M) / Training time (s) | FB15K-DB15K | | | FB15K-YAGO15K | | |
|---|---|---|---|---|---|---|---|
| | | Hits@1↑ | Hits@10↑ | MRR↑ | Hits@1↑ | Hits@10↑ | MRR↑ |
| EVA | 10.2/1,467.6 | .199 | .448 | .283 | .153 | .361 | .224 |
| MSNEA | 11.5/775.2 | .114 | .296 | .175 | .103 | .249 | .153 |
| MCLEA | 13.2/285.4 | .295 | .582 | .393 | .254 | .484 | .332 |
| GEEA$_{\text{SMALL}}$ | 11.2/217.3 | .322 | .602 | .417 | .270 | .513 | .352 |
| GEEA | 13.9/252.4 | **.343** | **.661** | **.450** | **.298** | **.585** | **.393** |

Figure 3: MRR results on FBDB15K, w.r.t. epochs.

## 3.5 IMPLEMENTATION DETAILS

We take Figure 2 as an example to illustrate the workflow of GEEA. First, the sub-embeddings outputted by $\mathcal{M}$ are used as input for sub-VAEs (top-left). Then, the reconstructed sub-embeddings are passed to respective decoders to predict the concrete features of different modalities (top-right). The conventional entity alignment prediction loss is also retained in GEEA (bottom-left). The latent variables outputted by sub-VAEs are further used to match the predefined normal distribution (bottom-center). The reconstructed sub-embeddings are fed into the fusion layer to obtain a reconstructed joint embedding, which is used to match the true joint embedding for post reconstruction (bottom-right). The final training loss is defined as:

$$\mathcal{L} = \sum_{f \in \mathcal{F}} \Big( \underbrace{\sum_{m \in \{g,a,i,...\}} \mathcal{L}_{\mathbf{m}_f}}_{\text{prior reconstruction}} + \underbrace{\mathcal{L}_f}_{\text{post reconstruction}} \Big) + \underbrace{\sum_{m \in \{g,a,i,...\}} \mathcal{L}_{\text{kld},m}}_{\text{distribution matching term}} + \underbrace{\mathcal{L}_{\text{ns}}}_{\text{prediction matching term}} \tag{22}$$

$$\underbrace{\phantom{\sum_{f \in \mathcal{F}} \Big( \sum_{m} \mathcal{L}_{\mathbf{m}_f} + \mathcal{L}_f \Big)}}_{\text{reconstruction term}}$$

where $\mathcal{F} = \{x \to x, y \to y, x \to y, y \to x\}$ is the set of all flows, and $\{g, a, i, ...\}$ is the set of all available modalities. For more details, please refer to Appendix B.

## 4 EXPERIMENTS

### 4.1 SETTINGS

We used the multi-modal EEA benchmarks (DBP15K (Sun et al., 2017), FB15K-DB15K and FB15K-YAGO15K (Chen et al., 2020)) as datasets, excluding surface information (i.e., the textual label information) to prevent data leakage (Sun et al., 2020b; Chen et al., 2022b). The baselines MUGNN (Cao et al., 2019), AliNet (Sun et al., 2020a) and decentRL (Guo et al., 2020) are methods tailored to relational graphs, while EVA (Liu et al., 2021), MSNEA (Chen et al., 2022a) and MCLEA (Lin et al., 2022) are state-of-the-art multi-modal EEA methods. We chose MCLEA (Lin et al., 2022) as the EEA model of GEEA and NeoEA (Guo et al., 2022b) in the main experiments. The results of using other models (e.g., EVA and MSNEA) can be found in Appendix C. The neural layers and input/hidden/output dimensions were kept identical for fair comparison.

### 4.2 ENTITY ALIGNMENT RESULTS

The entity alignment results on DBP15K are shown in Tables 1. The multi-modal methods significantly outperformed the single-modal methods, demonstrating the strength of leveraging different

Table 3: Entity synthesis results on five datasets. PRE ($\times 10^{-2}$), RE ($\times 10^{-2}$) denote the reconstruction errors for prior concrete features and output embeddings, respectively.

| Models | DBP15K$_{\text{ZH-EN}}$ | | | DBP15K$_{\text{JA-EN}}$ | | | DBP15K$_{\text{FR-EN}}$ | | | FB15K-DB15K | | | FB15K-YAGO15K | | |
|---|---|---|---|---|---|---|---|---|---|---|---|---|---|---|---|
| | PRE↓ | RE↓ | FID↓ | PRE↓ | RE↓ | FID↓ | PRE↓ | RE↓ | FID↓ | PRE↓ | RE↓ | FID↓ | PRE↓ | RE↓ | FID↓ |
| MCLEA + decoder | 8.104 | 4.218 | N/A | 7.640 | 5.441 | N/A | 10.578 | 5.985 | N/A | 18.504 | inf | N/A | 20.997 | inf | N/A |
| VAE + decoder | 0.737 | 0.206 | 1.821 | 0.542 | 0.329 | 2.184 | 0.856 | 0.689 | 3.083 | 10.564 | 11.354 | 10.495 | 9.645 | 9.982 | 16.180 |
| Sub-VAEs + decoder | 0.701 | 0.246 | 1.920 | 0.531 | 0.291 | 2.483 | 0.514 | 0.663 | 2.694 | 3.557 | 15.589 | 4.340 | 2.424 | 5.576 | 5.503 |
| GEEA | **0.438** | **0.184** | **0.935** | **0.385** | **0.195** | **1.871** | **0.451** | **0.121** | **2.422** | **3.141** | **6.151** | **3.089** | **1.730** | **2.039** | **3.903** |

Figure 4: Entity alignment results on FBDB15K, w.r.t. ratios of training alignment.

resources. Remarkably, our GEEA achieved new state-of-the-art performance on all three datasets across all metrics. The superior performance empirically verified the correlations between the generative objectives and EEA objective. In Table 2, we compared the performance of the multi-modal methods on FB15K-DB15K and FB15K-YAGO15K, where GEEA remained the best-performing method. Nevertheless, we observe that GEEA had more parameters compared with others, as it used VAEs and decoders to decode the embeddings back to concrete features. To probe the effectiveness of GEEA, we reduced the number of neurons to construct a GEEA$_{\text{SMALL}}$ and it still outperformed others with a significant margin.

In Figure 3, we plotted the MRR results w.r.t. training epochs on FBDB15K, where MCLEA and GEEA learned much faster than the methods with fewer parameters (i.e., EVA and MSNEA). In Figure 4, we further compared the performance of these two best-performing methods under different ratios of training alignment. We can observe that our GEEA achieved consistent better performance than MCLEA across various settings and metrics. The performance gap was more significantly when there were fewer training entity alignments ($\leq 30\%$). For instance, GEEA surpassed the second-best method by 36.1% in Hits@1 when only 10% aligned entity pairs were used for training.

In summary, the primary weakness of GEEA is its higher parameter count compared to existing methods. However, we demonstrated that a compact version of GEEA still outperformed the baselines in Table 2. This suggests that its potential weakness is manageable. Additionally, GEEA excelled in utilizing training data, achieving greater performance gains with less available training data.

## 4.3 ENTITY SYNTHESIS RESULTS

We conducted entity synthesis experiments by modifying the EEA benchmarks. We randomly selected 30% of the source entities in the testing alignment set as *dangling entities*, and removed the information of their counterpart entities during training. The goal was to reconstruct the information of their counterpart entities. We evaluated the performance using several metrics: the prior reconstruction error (PRE) for concrete features, the reconstruction error (RE) for the sub-embeddings, and Frechet inception distance (FID) for unconditional synthesis (Heusel et al., 2017). FID is a popular metric for evaluating generative models by measuring the feature distance between real and generated samples.

We implemented several baselines for comparison and present the results in Table 3: MCLEA with the decoders performed worst and it could not generate new entities unconditionally. Using Sub-VAEs to process different modalities performed better than using one VAE to process all modalities. However, the VAEs in Sub-VAEs could not support each other, and sometimes they failed to reconstruct the embeddings (e.g., the RE results on FB15K-DB15K). By contrast, our GEEA consistently and significantly outperformed these baselines. We also noticed that the results on FB15K-DB15K and FB15K-YAGO15K were worse than those on DBP15K. This could be due to the larger heterogeneity between two KGs compared to the heterogeneity between two languages of the same KG.

We present some generated samples of GEEA conditioned on the source dangling entities in Table 4. GEEA not only generated samples with the exact information that existed in the target KG, but also completed the target entities with highly reliable predictions. For example, the entity *Star Wars (film)*

Table 4: Entity synthesis samples from the FB15K-DB15K dataset. The **boldfaced** denotes the exactly matched entry, while the underlined denotes the potentially true entry.

| Source | | Target | | | GEEA Output | | |
|---|---|---|---|---|---|---|---|
| Entity | Image | Image | Neighborhood | Attribute | Image | Neighborhood | Attribute |
| Star Wars (film) | | | 20th Century Fox, George Lucas, John Williams | runtime, gross, budget | | **20th Century Fox**, **George Lucas**, Star Wars: Episode II, Willow (film), Aliens (film), Star Wars: The Clone War | initial release date, **runtime**, **budget**, **gross**, imdbId, numberOfEpisodes |
| George Harrison (musician) | | | The Beatles, Guitar, Rock music, Klaus Voormann, Jeff Lynne, Pop music | birthDate, deathDate, activeYearsStartYear, activeYearsEndYear, imdbId | | **The Beatles**, The Band, Ringo Starr, **Klaus Voormann**, **Jeff Lynne**, **Rock music** | deathYear, birthYear, **deathDate**, **birthDate**, **activeYearsStartYear**, **activeYearsEndYear**, **imdbId**, height, networth |

Table 5: Ablation study results on DBP15K$_{\text{ZH-EN}}$.

| Prediction Match | Distribution Match | Prior Reconstruction | Post Reconstruction | Entity Alignment | | | Entity Synthesis | | |
|---|---|---|---|---|---|---|---|---|---|
| | | | | Hits@1↑ | Hits@10↑ | MRR↑ | PRE↓ | RE↓ | FID↓ |
| √ | √ | √ | √ | **.761** | **.946** | **.827** | **0.438** | **0.184** | **0.935** |
| | √ | √ | √ | .045 | .186 | .095 | 0.717 | 0.306 | 2.149 |
| √ | | √ | √ | .702 | .932 | .783 | 0.551 | 0.193 | 1.821 |
| √ | √ | | √ | .746 | .930 | .813 | inf | 0.267 | 1.148 |
| √ | √ | √ | | .750 | .942 | .819 | 0.701 | 0.246 | 1.920 |

in target KG only had three basic attributes in the target KG, but GEEA predicted that it may also have the attributes like *imdbid* and *initial release data*.

## 4.4 ABLATION STUDY

We conducted ablation studies to verify the effectiveness of each module in GEEA. In Table 5, we can observe that the best results were achieved by the complete GEEA, and removing any module resulted in a performance loss. Interestingly, GEEA still worked even if we did not employ an EEA loss (the 2nd row) in the entity alignment experiment. It captured alignment information without the explicit optimization of the entity alignment objective through contrastive loss, which is an indispensable module in previous EEA methods. This observation further validates the effectiveness of GEEA.

## 5 RELATED WORKS

**Embedding-based Entity Alignment**   Most pioneer works focus on modeling the relational graph information. They can be divided into triplet-based (Sun et al., 2017; Pei et al., 2019a) and GNN-based (Wang et al., 2018; Guo et al., 2020). Recent methods explore multi-modal KG embedding for EEA (Zhang et al., 2019; Chen et al., 2022b; Lin et al., 2022). Although GEEA is designed for multi-modal EEA, it differs by focusing on objective optimization rather than specific models. GAN-based methods (Pei et al., 2019a; Guo et al., 2022b) are closely related to GEEA but distinct, as GEEA prioritizes the reconstruction process, while the existing methods focus on processing relational graph information for EEA.

**Variational Autoencoder**   We draw the inspiration from various excellent works, e.g., VAEs, flow-based models, GANs, and diffusion models that have achieved state-of-the-art performance in many fields (Heusel et al., 2017; Kong et al., 2020; Mittal et al., 2021; Nichol & Dhariwal, 2021; Ho et al., 2020; Rombach et al., 2022). Furthermore, recent studies (Hoogeboom et al., 2022; Li et al., 2022) find that these generative models can be used in controllable text generation. To the best of our knowledge, GEEA is the first method capable of generating new entities with concrete features. The design of M-VAE, prior and post reconstruction also differs from existing generative models and may offer insights for other domains.

## 6 CONCLUSION

This paper presents a theoretical analysis of how generative models can enhance EEA learning and introduces GEEA to address the limitations of existing GAN-based methods. Experiments demonstrate that GEEA achieves state-of-the-art performance in entity alignment and entity synthesis tasks. Future work will focus on designing new multi-modal encoders to enhance generative ability.

## ACKNOWLEDGMENT

We would like to thank all anonymous reviewers for their insightful and invaluable comments. This work is funded by National Natural Science Foundation of China (NSFCU23B2055/NSFCU19B2027/NSFC91846204), Zhejiang Provincial Natural Science Foundation of China (No.LGG22F030011), Fundamental Research Funds for the Central Universities (226-2023-00138), and the EPSRC project ConCur (EP/V050869/1).

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

# A  PROOFS OF THINGS

## A.1  THE COMPLETE PROOF OF PROPOSITION 1

*Proof.* Let $x \sim \mathcal{X}, y \sim \mathcal{Y}$ be two entities sampled from the entity sets $\mathcal{X}, \mathcal{Y}$, respectively. The main target of EEA is to learn a predictor that estimates the conditional probability $p_\theta(\mathbf{x}|\mathbf{y})$ (and reversely $p_\theta(\mathbf{y}|\mathbf{x})$), where $\theta$ represents the parameter set. For simplicity, we assume that the reverse function $p_\theta(\mathbf{y}|\mathbf{x})$ shares the same parameter set with $p_\theta(\mathbf{x}|\mathbf{y})$.

Now, suppose that one wants to learn a generative model for generating entity embeddings:

$$\log p(\mathbf{x}) = \log p(\mathbf{x}) \int p_\theta(\mathbf{y}|\mathbf{x}) d\mathbf{y} \tag{23}$$

$$= \int p_\theta(\mathbf{y}|\mathbf{x}) \log p(\mathbf{x}) d\mathbf{y} \tag{24}$$

$$= \mathbb{E}_{p_\theta(\mathbf{y}|\mathbf{x})}[\log p(\mathbf{x})] \tag{25}$$

$$= \mathbb{E}_{p_\theta(\mathbf{y}|\mathbf{x})}\left[\log \frac{p(\mathbf{x}, \mathbf{y})}{p(\mathbf{y}|\mathbf{x})}\right] \tag{26}$$

$$= \mathbb{E}_{p_\theta(\mathbf{y}|\mathbf{x})}\left[\log \frac{p(\mathbf{x}, \mathbf{y})p_\theta(\mathbf{y}|\mathbf{x})}{p(\mathbf{y}|\mathbf{x})p_\theta(\mathbf{y}|\mathbf{x})}\right] \tag{27}$$

$$= \mathbb{E}_{p_\theta(\mathbf{y}|\mathbf{x})}\left[\log \frac{p(\mathbf{x}, \mathbf{y})}{p_\theta(\mathbf{y}|\mathbf{x})}\right] + \mathbb{E}_{p_\theta(\mathbf{y}|\mathbf{x})}\left[\log \frac{p_\theta(\mathbf{y}|\mathbf{x})}{p(\mathbf{y}|\mathbf{x})}\right] \tag{28}$$

$$= \mathbb{E}_{p_\theta(\mathbf{y}|\mathbf{x})}\left[\log \frac{p(\mathbf{x}, \mathbf{y})}{p_\theta(\mathbf{y}|\mathbf{x})}\right] + D_{\mathrm{KL}}(p_\theta(\mathbf{y}|\mathbf{x}) \parallel p(\mathbf{y}|\mathbf{x})), \tag{29}$$

where the left-hand side of Equation (7) is the evidence lower bound (ELBO) (Kingma & Welling, 2013), and the right-hand side is the KL divergence (Kullback & Leibler, 1951) between our parameterized distribution $p_\theta(\mathbf{y}|\mathbf{x})$ (i.e., the predictor) and the true distribution $p(\mathbf{y}|\mathbf{x})$.

The recent GAN-based methods (Pei et al., 2019a;b; Guo et al., 2022b) propose to leverage the entities out of training set for unsupervised learning. Their common idea is to make the entity embeddings from different KGs indiscriminative to a discriminator, and the underlying aligned entities shall be encoded in the same way and have similar embeddings. To formally prove this idea, we dissect the ELBO in Equation (7) as follows:

$$\mathbb{E}_{p_\theta(\mathbf{y}|\mathbf{x})}\left[\log \frac{p(\mathbf{x}, \mathbf{y})}{p_\theta(\mathbf{y}|\mathbf{x})}\right] = \mathbb{E}_{p_\theta(\mathbf{y}|\mathbf{x})}\left[\log \frac{p_\theta(\mathbf{x}|\mathbf{y})p(\mathbf{y})}{p_\theta(\mathbf{y}|\mathbf{x})}\right] \tag{30}$$

$$= \mathbb{E}_{p_\theta(\mathbf{y}|\mathbf{x})}\left[\log p_\theta(\mathbf{x}|\mathbf{y})\right] + \mathbb{E}_{p_\theta(\mathbf{y}|\mathbf{x})}\left[\log \frac{p(\mathbf{y})}{p_\theta(\mathbf{y}|\mathbf{x})}\right] \tag{31}$$

$$= \mathbb{E}_{p_\theta(\mathbf{y}|\mathbf{x})}\left[\log p_\theta(\mathbf{x}|\mathbf{y})\right] - D_{\mathrm{KL}}(p_\theta(\mathbf{y}|\mathbf{x}) \parallel p(\mathbf{y})) \tag{32}$$

Therefore, we have:

$$\log p(\mathbf{x}) = \underbrace{\mathbb{E}_{p_\theta(\mathbf{y}|\mathbf{x})}\left[\log p_\theta(\mathbf{x}|\mathbf{y})\right]}_{\text{reconstruction term}} - \underbrace{D_{\mathrm{KL}}(p_\theta(\mathbf{y}|\mathbf{x}) \parallel p(\mathbf{y}))}_{\text{distribution matching term}} + \underbrace{D_{\mathrm{KL}}(p_\theta(\mathbf{y}|\mathbf{x}) \parallel p(\mathbf{y}|\mathbf{x}))}_{\text{prediction matching term}} \tag{33}$$

The first term aims to reconstruct the original embedding $\mathbf{x}$ based on $\mathbf{y}$ generated from $\mathbf{x}$, which has not been studied by the existing discriminative EEA methods (Guo et al., 2020; Liu et al., 2021; Lin et al., 2022). The second term imposes the distribution $\mathbf{y}$ conditioned on $\mathbf{x}$ to match the prior distribution of $\mathbf{y}$, which has been investigated by the GAN-based EEA methods (Pei et al., 2019a;b; Guo et al., 2022b). The third term is the main objective of EEA (i.e., Equation (8) with the target $p(\mathbf{y}|\mathbf{x})$ being only partially observed).

Note that, $p(\mathbf{x})$ is irrelevant to our parameter set $\theta$ and thus can be regarded as a constant during optimization. Therefore, maximizing the ELBO (i.e., maximizing the first term and minimizing the second term) will result in minimizing the third term, concluding the proof. □

## A.2 PROOF OF PROPOSITION 2

*Proof.* We first have a look on the right hand:

$$D_{\mathrm{KL}}(p(\mathbf{z}_{x\to x}), p(\mathbf{z}^*)) + D_{\mathrm{KL}}(p(\mathbf{z}_{y\to y}), p(\mathbf{z}^*)) \tag{34}$$

Ideally, all the variables $\mathbf{z}_{x\to x}$, $\mathbf{z}_{y\to y}$, and $\mathbf{z}^*$ follow the Gaussian distributions with $\mu_{x\to x}$, $\mu_{y\to y}$, $\mu^*$ and $\sigma_{x\to x}$, $\sigma_{y\to y}$, $\sigma^*$ as mean and variance, respectively.

Luckily, we can use the following equation to calculate the KL divergence between two Gaussian distributions conveniently:

$$D_{\mathrm{KL}}(p(\mathbf{z}_1), p(\mathbf{z}_2)) = \log \frac{\sigma_2}{\sigma_1} + \frac{\sigma_1^2 + (\mu_1 - \mu_2)^2}{2\sigma_2^2} - \frac{1}{2}, \tag{35}$$

and rewrite Equation(34) as:

$$D_{\mathrm{KL}}(p(\mathbf{z}_{x\to x}), p(\mathbf{z}^*)) + D_{\mathrm{KL}}(p(\mathbf{z}_{y\to y}), p(\mathbf{z}^*)) \tag{36}$$

$$= (\log \frac{\sigma^*}{\sigma_{x\to x}} + \frac{\sigma_{x\to x}^2 + (\mu_{x\to x} - \mu^*)^2}{2(\sigma^*)^2} - \frac{1}{2}) + (\log \frac{\sigma^*}{\sigma_{y\to y}} + \frac{\sigma_{y\to y}^2 + (\mu_{y\to y} - \mu^*)^2}{2(\sigma^*)^2} - \frac{1}{2}) \tag{37}$$

$$= (\log \frac{\sigma^*}{\sigma_{x\to x}} + \log \frac{\sigma^*}{\sigma_{y\to y}}) + (\frac{\sigma_{x\to x}^2 + (\mu_{x\to x} - \mu^*)^2}{2(\sigma^*)^2} + \frac{\sigma_{y\to y}^2 + (\mu_{y\to y} - \mu^*)^2}{2(\sigma^*)^2}) - 1 \tag{38}$$

$$= (\log \frac{\sigma^*}{\sigma_{x\to x}} + \log \frac{\sigma^*}{\sigma_{y\to y}}) + \frac{\sigma_{x\to x}^2 + (\mu_{x\to x} - \mu^*)^2 + \sigma_{y\to y}^2 + (\mu_{y\to y} - \mu^*)^2}{2(\sigma^*)^2} - 1 \tag{39}$$

Take $\mathbf{z}^* \sim \mathcal{N}(\mu^* = \mathbf{0}, \sigma^* = \mathbf{I})$ into the above equation, we will have:

$$D_{\mathrm{KL}}(p(\mathbf{z}_{x\to x}), p(\mathbf{z}^*)) + D_{\mathrm{KL}}(p(\mathbf{z}_{y\to y}), p(\mathbf{z}^*)) \tag{40}$$

$$= -\log \sigma_{x\to x}\sigma_{y\to y} + \frac{1}{2}(\sigma_{x\to x}^2 + \sigma_{y\to y}^2 + \mu_{x\to x}^2 + \mu_{y\to y}^2) - 1 \tag{41}$$

Similarly, the left hand can be expanded as:

$$D_{\mathrm{KL}}(p(\mathbf{z}_{x\to x}), p(\mathbf{z}_{y\to y})) = \log \frac{\sigma_{y\to y}}{\sigma_{x\to x}} + \frac{\sigma_{x\to x}^2 + (\mu_{x\to x} - \mu_{y\to y})^2}{2\sigma_{y\to y}^2} - \frac{1}{2}, \tag{42}$$

Thus, the difference between the left hand and the right hand can be computed:

$$D_{\mathrm{KL}}(p(\mathbf{z}_{x\to x}), p(\mathbf{z}^*)) + D_{\mathrm{KL}}(p(\mathbf{z}_{y\to y}), p(\mathbf{z}^*)) - D_{\mathrm{KL}}(p(\mathbf{z}_{x\to x}), p(\mathbf{z}_{y\to y})) \tag{43}$$

$$= -\log \sigma_{x\to x}\sigma_{y\to y} + \frac{1}{2}(\sigma_{x\to x}^2 + \sigma_{y\to y}^2 + \mu_{x\to x}^2 + \mu_{y\to y}^2) - 1 \tag{44}$$

$$- (\log \frac{\sigma_{y\to y}}{\sigma_{x\to x}} + \frac{\sigma_{x\to x}^2 + (\mu_{x\to x} - \mu_{y\to y})^2}{2\sigma_{y\to y}^2} - \frac{1}{2}) \tag{45}$$

$$= (-\log \sigma_{x\to x}\sigma_{y\to y} - \log \frac{\sigma_{y\to y}}{\sigma_{x\to x}}) \tag{46}$$

$$+ (\frac{1}{2}(\sigma_{x\to x}^2 + \sigma_{y\to y}^2 + \mu_{x\to x}^2 + \mu_{y\to y}^2) - \frac{\sigma_{x\to x}^2 + (\mu_{x\to x} - \mu_{y\to y})^2}{2\sigma_{y\to y}^2}) + (-1 + \frac{1}{2}) \tag{47}$$

$$= -2\log \sigma_{y\to y} - \frac{1}{2} \tag{48}$$

$$+ \frac{\sigma_{x\to x}^2\sigma_{y\to y}^2 + \sigma_{y\to y}^4 + \mu_{x\to x}^2\sigma_{y\to y}^2 + \mu_{y\to y}^2\sigma_{y\to y}^2 - \sigma_{x\to x}^2 - \mu_{x\to x}^2 - \mu_{y\to y}^2 + 2\mu_{x\to x}\mu_{y\to y}}{2\sigma_{y\to y}^2} \tag{49}$$

$$= -2\log \sigma_{y\to y} - \frac{1}{2} \tag{50}$$

$$+ \frac{(\sigma_{y\to y}^2 - 1)\sigma_{x\to x}^2 + (\mu_{x\to x}^2 + \mu_{y\to y}^2)(\sigma_{y\to y}^2 - 1) + \sigma_{y\to y}^4 + 2\mu_{x\to x}\mu_{y\to y}}{2\sigma_{y\to y}^2} \tag{51}$$

$$= -2\log \sigma_{y\to y} - \frac{1}{2} + \frac{(\mu_{x\to x}^2 + \mu_{y\to y}^2 + \sigma_{x\to x}^2)(\sigma_{y\to y}^2 - 1) + \sigma_{y\to y}^4 + 2\mu_{x\to x}\mu_{y\to y}}{2\sigma_{y\to y}^2} \tag{52}$$

As we optimize $\mathbf{z}_{y\to y} \to \mathbf{z}^*$, i.e., minimize $D_{\mathrm{KL}}(p(\mathbf{z}_{y\to y}), p(\mathbf{z}^*))$, we will have:

$$\log \sigma_{y\to y} \to 0, \quad \sigma_{y\to y}^2 - 1 \to 0, \quad \mu_{x\to x}\mu_{y\to y} \to 0, \quad \sigma_{y\to y}^4 \to 1, \tag{53}$$

and consequently:

$$D_{\mathrm{KL}}(p(\mathbf{z}_{x\to x}), p(\mathbf{z}^*)) + D_{\mathrm{KL}}(p(\mathbf{z}_{y\to y}), p(\mathbf{z}^*)) - D_{\mathrm{KL}}(p(\mathbf{z}_{x\to x}), p(\mathbf{z}_{y\to y})) \to 0, \tag{54}$$

Similarly, as we optimize $\mathbf{z}_{x\to x} \to \mathbf{z}^*$, i.e., minimize $D_{\mathrm{KL}}(p(\mathbf{z}_{x\to x}), p(\mathbf{z}^*))$, we will have:

$$D_{\mathrm{KL}}(p(\mathbf{z}_{x\to x}), p(\mathbf{z}^*)) + D_{\mathrm{KL}}(p(\mathbf{z}_{y\to y}), p(\mathbf{z}^*)) - D_{\mathrm{KL}}(p(\mathbf{z}_{y\to y}), p(\mathbf{z}_{x\to x})) \to 0 \tag{55}$$

Therefore, jointly minimizing $D_{\mathrm{KL}}(p(\mathbf{z}_{x\to x}), p(\mathbf{z}^*))$ and $D_{\mathrm{KL}}(p(\mathbf{z}_{y\to y}), p(\mathbf{z}^*))$ will subsequently minimizing $D_{\mathrm{KL}}(p(\mathbf{z}_{x\to x}), p(\mathbf{z}_{y\to y}))$ and $D_{\mathrm{KL}}(p(\mathbf{z}_{y\to y}), p(\mathbf{z}_{x\to x}))$, and finally aligning the distributions between $\mathbf{x}$ and $\mathbf{y}$, concluding the proof. $\qquad\square$

# B  IMPLEMENTATION DETAILS

## B.1  DECODING EMBEDDINGS BACK TO CONCRETE FEATURES

All decoders used to decode the reconstructed embeddings to the concrete features comprise several hidden layers and an output layer. Specifically, each hidden layer has a linear layer with layer norm and ReLU/Tanh activations. The output layer is different for different modalities. For the relational graph and attribute information, their concrete features are organized in the form of multi-classification labels. For example, the relational graph information $g_i$ for an entity $x_i$ is represented by:

$$g_i = (0, ..., 1, ...1, ..., 0)^T, \quad |g_i| = |\mathcal{X}|, \tag{56}$$

where $g_i$ has $|\mathcal{X}|$ elements with 1 indicating the connection and 0 otherwise. Therefore, the output layer transforms the hidden output to the concrete feature prediction with a matrix $W_o \in \mathbb{R}^{H \times |\mathcal{X}|}$, where $H$ is the output dimension of the final hidden layer.

The image concrete features are actually the pretrained embeddings rather than pixel data, as we use the existing EEA models for embedding entities. Therefore, we replaced the binary cross-entropy loss with a MSE loss to train GEEA to recover this pretrained embedding.

## B.2  IMPLEMENTING A GEEA

We implement GEEA with PyTorch and run the main experiments on a RTX 4090. We illustrate the training procedure of GEEA as outlined in Algorithm 1. We first initialize all trainable variables and the get the mini-batch data of supervised flows $x \to y$, $y \to x$ and unsupervised flows $x \to x$, $y \to y$, respectively.

For the supervised flows, we iterate the batched data and calculate the prediction matching loss which is also used in most existing works. Then, we calculate the distribution matching, prior reconstruction and post reconstruction losses and sum them for later joint optimization.

For the unsupervised flows, we first process the raw feature with $\mathcal{M}$ and *VAE* to obtain the embeddings and reconstructed embeddings. Then we estimate the distribution matching loss with the embedding sets as input (Equation (16)), after which we calculate the prior and post reconstruction loss for each $x$ and each $y$.

Finally, we sum all the losses produced with all flows, and minimize them until the performance on the valid dataset does not improve.

The overall hyper-parameter settings in the main experiments are presented in Table 6.

# C  ADDITIONAL EXPERIMENTS

## C.1  DATASETS

We present the statistics of entity alignment and entity synthesis datasets in Table 7. To construct an entity synthesis dataset, we first sample 30% of entity alignments from the testing set of the original

---

**Algorithm 1** Generative Embedding-based Entity Alignment

---

1: **Input:** The entity sets $\mathcal{X}$, $\mathcal{Y}$, the multi-modal information $\mathcal{G}$, $\mathcal{A}$, $\mathcal{I}$ ..., the EEA model $\mathcal{M}$, and M-VAE *VAE*;
2: Randomly initialize all parameters;
3: **repeat**
4:    $\mathcal{B}_{\text{sup}} \leftarrow \{(x,y)|(x,y) \sim \mathcal{S}\}$;    *// get a batch of supervised training data*
5:    $\mathcal{B}_{\text{unsup}} \leftarrow \{(x,y)|x \sim \mathcal{X}, y \sim \mathcal{Y}\}$;    *// get a batch of unsupervised training data*
6:    **for** $(x,y) \in \mathcal{B}_{\text{sup}}$ **do**
7:       $\mathbf{x}, \mathbf{y} \leftarrow \mathcal{M}(x), \mathcal{M}(y)$;    *// obtain embeddings and sub-embeddings*
8:       $\mathbf{y}_{x \rightarrow y}, \mathbf{x}_{y \rightarrow x} \leftarrow VAE(\mathbf{x}), VAE(\mathbf{y})$;    *// obtain the reconstructed mutual embeddings*
9:       Calculate the prediction matching loss following Equation (8);
10:      Calculate the prior reconstruction loss following Equation (19);
11:      Calculate the post reconstruction loss following Equation (20);
12:    **end for**
13:   $\{\mathbf{x}_{x \rightarrow x}, \mathbf{y}_{y \rightarrow y}|(x,y) \in \mathcal{B}_{\text{unsup}}\} \leftarrow \{VAE(\mathcal{M}(x)), VAE(\mathcal{M}(y))|(x,y) \in \mathcal{B}_{\text{unsup}}\}$; *// obtain the reconstructed self embeddings*
14:    Calculate the distribution matching loss following Equation (16);
15:    Calculate the prior reconstruction loss following Equation (19);
16:    Calculate the post reconstruction loss following Equation (20);
17:    Jointly minimize all losses;
18: **until** the performance does not improve.

---

Table 6: Hyper-parameter settings in the main experiments. PM,DM, PrioR, PostR denote prediction matching, distribution matching, prior reconstruction, and post reconstruction, respectively.

| Datasets | # epoch | batch-size | # VAE layers | learning rate | optimizer | dropout rate | unsupervised batch-size | flow weights (xx,yy,xy,yx) | loss weights (PM,DM, PrioR, PostR) | hidden sizes | latent size | decoder hidden sizes |
|---|---|---|---|---|---|---|---|---|---|---|---|---|
| DBP15K$_{\text{ZH-EN}}$ | 200 | 2,500 | 2 | 0.001 | Adam | 0.5 | 2,800 | [1.,1.,5.,5.] | [1., 0.5,1.,1.] | [300,300] | 300 | [300,1000] |
| DBP15K$_{\text{JA-EN}}$ | 200 | 2,500 | 2 | 0.001 | Adam | 0.5 | 2,800 | [1.,1.,5.,5.] | [1., 0.5,1.,1.] | [300,300] | 300 | [300,1000] |
| DBP15K$_{\text{FR-EN}}$ | 200 | 2,500 | 2 | 0.001 | Adam | 0.5 | 2,800 | [1.,1.,5.,5.] | [1., 0.5,1.,1.] | [300,300] | 300 | [300,1000] |
| FB15K-DB15K | 300 | 3,500 | 3 | 0.0005 | Adam | 0.5 | 2,500 | [1.,1.,5.,5.] | [1., 0.5,1.,1.] | [300,300,300] | 300 | [300,300,1000] |
| FB15K-YAGO15K | 300 | 3,500 | 3 | 0.0005 | Adam | 0.5 | 2,500 | [1.,1.,5.,5.] | [1., 0.5,1.,1.] | [300,300,300] | 300 | [300,300,1000] |

Table 7: Statistics of the datasets.

| Datasets | Entity Alignment | Entity Synthesis | | # Entities | # Relations | # Attributes | # Images |
|---|---|---|---|---|---|---|---|
| | # Test Alignments | # Known Test Alignments | # Unknown Test Alignments | | | | |
| DBP15K$_{\text{ZH-EN}}$ | 10,500 | 7,350 | 3,150 | 19,388 | 1,701 | 8,111 | 15,912 |
| | 10,500 | 7,350 | 3,150 | 19,572 | 1,323 | 7,173 | 14,125 |
| DBP15K$_{\text{JA-EN}}$ | 10,500 | 7,350 | 3,150 | 19,814 | 1,299 | 5,882 | 12,739 |
| | 10,500 | 7,350 | 3,150 | 19,780 | 1,153 | 6,066 | 13,741 |
| DBP15K$_{\text{FR-EN}}$ | 10,500 | 7,350 | 3,150 | 19,661 | 903 | 4,547 | 14,174 |
| | 10,500 | 7,350 | 3,150 | 19,993 | 1,208 | 6,422 | 13,858 |
| FB15K-DB15K | 10,276 | 7,193 | 3,083 | 14,951 | 1,345 | 116 | 13,444 |
| | 10,500 | 7,350 | 3,150 | 12,842 | 279 | 225 | 12,837 |
| FB15K-YAGO15K | 8,959 | 6,272 | 2,687 | 14,951 | 1,345 | 116 | 13,444 |
| | 10,500 | 7,350 | 3,150 | 15,404 | 32 | 7 | 11,194 |

entity alignment dataset. Then, we view the source entities in sampled entities pairs as the dangling entities, and make their target entities unseen during training. To this end, we remove all types of information referred to these target entities from the training set.

## C.2 SINGLE-MODAL GEEA

We first remove the image encoder from multi-modal EEA models. The results are shown in Table 11. Notably, our GEEA without the image encoder still achieves state-of-the-art performance on several metrics, such as Hits@10.

Then, we conducted new experiments on the OpenEA 100K (Sun et al., 2020b). Although OpenEA 100K does not have a multi-modal version, it is still interesting to explore the performance of GEEA with single-modal EEA models on it, similar to NeoEA (Guo et al., 2022b). We conducted experiments following the NeoEA and present the results in Table 12. It is clear that our method can

Table 8: Entity alignment results of GEEA with different EEA models on DBP15K datasets.

| Models | DBP15K_ZH-EN | | | DBP15K_JA-EN | | | DBP15K_FR-EN | | |
|---|---|---|---|---|---|---|---|---|---|
| | Hits@1↑ | Hits@10↑ | MRR↑ | Hits@1↑ | Hits@10↑ | MRR↑ | Hits@1↑ | Hits@10↑ | MRR↑ |
| EVA (Liu et al., 2021) | .680 | .910 | .762 | .673 | .908 | .757 | .683 | .923 | .767 |
| GEEA w/ EVA | **.715** | **.922** | **.794** | **.707** | **.925** | **.791** | **.727** | **.940** | **.817** |
| MSNEA (Chen et al., 2022a) | .601 | .830 | .684 | .535 | .775 | .617 | .543 | .801 | .630 |
| GEEA w/ MSNEA | **.643** | **.872** | **.732** | **.559** | **.821** | **.671** | **.586** | **.853** | **.672** |
| MCLEA (Lin et al., 2022) | .715 | .923 | .788 | .715 | .909 | .785 | .711 | .909 | .782 |
| GEEA w/ MCLEA | **.761** | **.946** | **.827** | **.755** | **.953** | **.827** | **.776** | **.962** | **.844** |

Table 9: More entity synthesis samples from different dataset. The first two rows are from FB15K-YAGO15K; the middle two rows are from DBP15K_ZH-EN; the last two rows are from DBP15K_FR-EN.

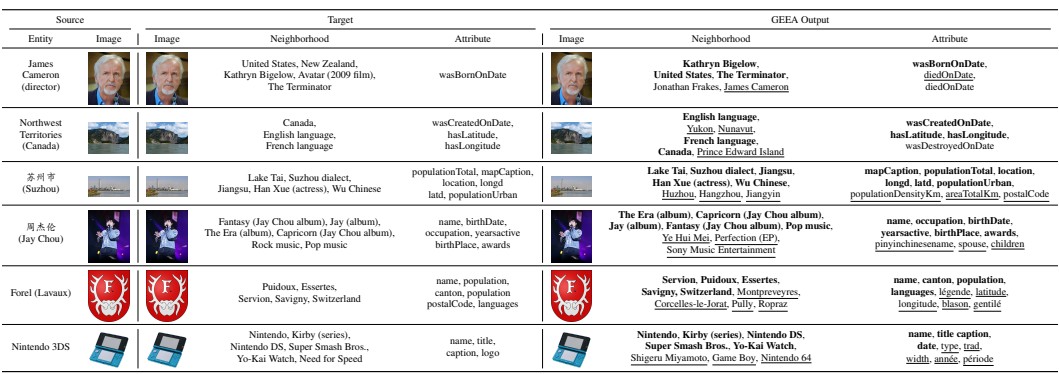

Table 10: Some false samples from FB15K-YAGO15K.

significantly enhance the performance of SEA (Pei et al., 2019a), which can be attributed to the more stringent objectives analyzed in Section 2.

## C.3 GEEA ON DANGLING ENTITY DETECTION

The methods for detecting dangling entities (Sun et al., 2021) can be categorized into three groups: (1) Nearest Neighbor Classifier (NNC), which trains a classifier to determine whether a source entity lacks a counterpart entity in the target KG; (2) Margin-based Ranking (MR), which learns a margin value $\lambda$. If the embedding distance between a source entity and its nearest neighbor in the target KG exceeds $\lambda$, this source entity is considered a dangling entity; (3) Background Ranking (BR), which regards the dangling entities as background and randomly pushes them away from aligned entities.

All three types of dangling detection methods heavily rely on the quality of entity embeddings. Therefore, if the proposed GEEA learns better embeddings for entity alignment, it is expected to contribute to the detection of dangling entities. To verify this idea, we conducted experiments on new datasets, following [1]. We used the same parameter settings and employed MTransE (Chen et al., 2017) as the backbone model. The results are presented in Table 13. Clearly, incorporating GEEA led to significant performance improvements in dangling entity detection across all three datasets. The performance gains were particularly notable in terms of precision and F1 metrics.

## C.4 GEEA WITH DIFFERENT EEA MODELS

We also investigated the performance of GEEA with different EEA models. As shown in Table 8, GEEA significantly improved all the baseline models on all metrics and datasets. Remarkably, the

Table 11: Detailed entity alignment results on DBP15K datasets, without surface information and iterative strategy.

| Models | DBP15K$_{\text{ZH-EN}}$ | | | DBP15K$_{\text{JA-EN}}$ | | | DBP15K$_{\text{FR-EN}}$ | | |
|---|---|---|---|---|---|---|---|---|---|
| | Hits@1↑ | Hits@10↑ | MRR↑ | Hits@1↑ | Hits@10↑ | MRR↑ | Hits@1↑ | Hits@10↑ | MRR↑ |
| EVA (Liu et al., 2021) | .680 | .910 | .762 | .673 | .908 | .757 | .683 | .923 | .767 |
| MSNEA (Chen et al., 2022a) | .601 | .830 | .684 | .535 | .775 | .617 | .543 | .801 | .630 |
| MCLEA (Lin et al., 2022) | .715 | .923 | .788 | .715 | .909 | .785 | .711 | .909 | .782 |
| MCLEA w/o image | .658 | .915 | .726 | .662 | .904 | .740 | .662 | .902 | .747 |
| GEEA | **.761** | **.946** | **.827** | **.755** | **.953** | **.827** | **.776** | **.962** | **.844** |
| GEEA w/o image | .709 | .929 | .782 | .708 | .935 | .784 | .717 | .946 | .796 |

Table 12: Single-modal entity alignment results on OpenEA 100K datasets.

| Models | EN-FR | | EN-DE | | DBPedia-WikiData | | DBPedia-Yago | |
|---|---|---|---|---|---|---|---|---|
| | Hits@1↑ | MRR↑ | Hits@1↑ | MRR↑ | Hits@1↑ | MRR↑ | Hits@1↑ | MRR↑ |
| SEA (Pei et al., 2019a) | .225 | .314 | .341 | .421 | .291 | .378 | .490 | .578 |
| NeoEA (SEA) (Guo et al., 2022b) | .254 | .345 | .364 | .446 | .325 | .416 | .569 | .651 |
| GEEA (SEA) | **.269** | **.355** | **.377** | **.459** | **.349** | **.436** | **.597** | **.685** |

performance of EVA with GEEA on some datasets like DBP15K$_{\text{FR-EN}}$ were even better than that of the original MCLEA.

## C.5 RESULTS WITH DIFFERENT ALIGNMENT RATIOS ON ALL DATASETS

We present the results with different alignment ratios on all datasets in Figure 5, which demonstrate the same conclusion as in Figure 4.

## C.6 MORE ENTITY SYNTHESIS SAMPLES

We illustrate more entity synthesis samples in Table 9 and some false samples in Table 10. The main reason for less accurate synthesis results is the lack of information. For example, in the FB15K-YAGO15K datasets, the YAGO KG has only 7 different attributes. Also, as some entities do not have image features, the EEA models are configured to initialize the pretrained image embeddings with random vectors. To mitigate this problem, we plan to design new benchmarks and new EEA models to directly process and generate the raw data in future work.

Table 13: Dangling entity detection results on DBP2.0.

| Models | DBP 2.0$_{ZH-EN}$ | | | DBP 2.0$_{JA-EN}$ | | | DBP 2.0$_{FR-EN}$ | | |
|---|---|---|---|---|---|---|---|---|---|
| | Precision↑ | Recall↑ | F1↑ | Precision↑ | Recall↑ | F1↑ | Precision↑ | Recall↑ | F1↑ |
| NNC (Sun et al., 2021) | .604 | .485 | .538 | .622 | .491 | .549 | .459 | .447 | .453 |
| GEEA (NNC) | .617 | .509 | .558 | .637 | .460 | .534 | .479 | .449 | .464 |
| MR (Sun et al., 2021) | .781 | .702 | .740 | .799 | .708 | .751 | .482 | .575 | .524 |
| GEEA (MR) | .793 | .709 | .749 | .812 | .714 | .760 | .508 | .594 | .548 |
| BR (Sun et al., 2021) | .811 | **.728** | .767 | .816 | .733 | .772 | .539 | .686 | .604 |
| GEEA (BR) | **.821** | .724 | **.769** | **.833** | **.735** | **.781** | **.549** | **.694** | **.613** |

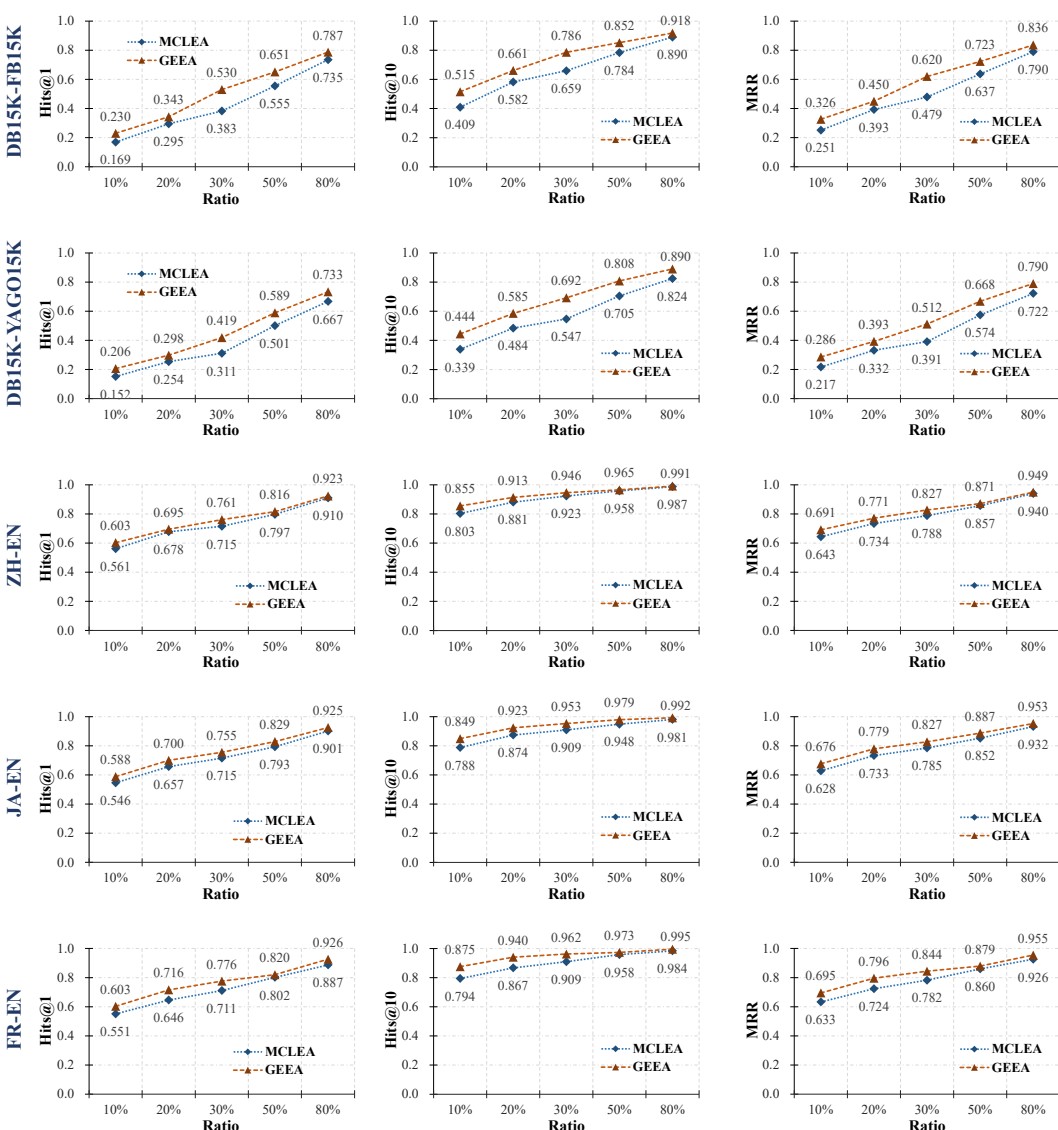

Figure 5: Entity alignment results on all datasets, w.r.t. ratios of training alignment.

