# OpenReview forum: "Revisit and Outstrip Entity Alignment: A Perspective of Generative Models"
_ICLR.cc/2024/Conference — ICLR 2024 poster_

### Official Review · Reviewer_sPPq · 2023-10-18

**Soundness:** 3 good
**Presentation:** 3 good
**Contribution:** 4 excellent
**Rating:** 8
**Confidence:** 4

**Summary:**

This paper presents a M-VAE based generative approach for embedding-based entity alignment. This paper theoretically justified the plausibility of formulating the problem as a generative task that is akin to what recent GAN-based EA approach has done. Specifically, for the problem with dangling entities, different from prvious work that has focused on detecting such dangling points, this work proposes a novel solution of entity synthesis. Experiments are done on common datasets shared by most prior works on this topic, with fair comparison ensured (by removing entity names, which has been violated by some prior works).

**Strengths:**

The contributions of this paper are from multiple perspectives. The plausibility of a generative formulation of the problem is theoretically justified. For the recently proposed dangling entity problem, a novel solution of entity synthesis is proposed to fulfill the missing targets in the target-side KG. Evaluation has covered the traditional setting of close-space entity alignment to show the effectiveness from that perspective, and a new, plausible setting is proposed for entity synthesis to show the effectiveness from this new angle of solution. The presented method and experiments look sound.

**Weaknesses:**

While the proposed entity synthesis approach leads to an essentially different solution to dangling entities from the previous dangling detection approaches, I wonder if the proposed approach can still can contribute to (and be compared with) dangling detection.
There is one detail that I might have missed: in the close-space EA setting without considering dangling entities, is there any technique of constrained generation/decoding that ensure the generation to fall into the candidate space?

**Questions:**

Please see weaknesses above.

---

> ### Author Response · Authors · 2023-11-13
>
> We are grateful for your encouraging comments and insightful observations. We hope the following responses address your concerns:
>
>
> ### Weaknesses:
>
> - **I wonder if the proposed approach can still contribute to (and be compared with) dangling detection.**
>
>     Great idea. The methods for detecting dangling entities can be categorized into three groups: (1) Nearest Neighbor Classifier (**NNC**), which trains a classifier to determine whether a source entity lacks a counterpart entity in the target KG; (2) Margin-based Ranking (**MR**), which learns a margin value $\lambda$. If the embedding distance between a source entity and its nearest neighbor in the target KG exceeds $\lambda$, this source entity is considered a dangling entity; (3) Background Ranking (**BR**), which regards the dangling entities as background and randomly pushes them away from aligned entities.
>
>     All three types of dangling detection methods heavily rely on the quality of entity embeddings. Therefore, if the proposed GEEA learns better embeddings for entity alignment, it is expected to contribute to the detection of dangling entities. To verify this idea, we conducted experiments on new datasets, following [1]. We used the same parameter settings and employed MTransE as the backbone model. The results are presented in the table below:
>
>     | Methods  | DBP 2.0 ZH-EN Precision | DBP 2.0 ZH-EN Recall | DBP 2.0 ZH-EN F1 | DBP 2.0 JA-EN Precision | DBP 2.0 JA-EN Recall | DBP 2.0 JA-EN F1 | DBP 2.0 FR-EN Precision | DBP 2.0 FR-EN Recall | DBP 2.0 FR-EN F1 |
>     |---|:---:|:---:|:---:|:---:|:---:|:---:|:---:|:---:|:---:|
>     | NNC  | .604 | .485 | .538 | .622 | .491 | .549 | .459 | .447 | .453 |
>     | GEEA (NNC)  | .617 | .509 | .558 | .637 | .460 | .534 | .479 | .449 | .464 |
>     | MR  | .781 | .702 | .740 | .799 | .708 | .751 | .482 | .575 | .524 |
>     | GEEA (MR)  | .793 | .709 | .749 | .812 | .714 | .760 | .508 | .594 | .548 |
>     | BR  | *.811* | **.728** | *.767* | *.816* | *.733* | *.772* | *.539* | *.686* | *.604* |
>     | GEEA (BR)  | **.821** | .724 | **.769** | **.833** | **.735** | **.781** | **.549** | **.694** | **.613** |
>
>     Clearly, incorporating GEEA leads to significant performance improvements in dangling entity detection across all three datasets. The performance gains are particularly notable in terms of precision and F1 metrics. We have added the dangling entity detection experiments in the revision.
>
> - **In the close-space EA setting without considering dangling entities, is there any technique of constrained generation/decoding that ensures the generation to fall into the candidate space?**
>
>     The reparameter trick of VAEs, prior reconstruction, and post reconstruction ensure the generation of new entities within the target space.
>
>     We apply the reparameter trick by adding a small noise vector to the latent embedding in VAEs. The latent embedding is then converted back to the input embedding. This process allows the model to learn the conversion between distributions rather than fixed features.
>
>     Recall that we consider four different reconstruction flows. Take x→y as an example, we use the entity alignment training set as supervised data, where each example is in the form of (x, y), representing a pair of aligned entities. The model takes the raw features of x from the source KG as input and reproduces the raw features of its counterpart y as output. We minimize the difference between the generated features and the true features of y as the prior reconstruction loss.
>
>     Additionally, we introduce post reconstruction. This involves reinputting the output embeddings of sub-VAEs to the fusion layer, resulting in a generated joint embedding. We then minimize the difference between this generated joint embedding and the joint embedding of y to ensure a consistent generation process across different modalities.
>
>
> [1] "Knowing the no-match: entity alignment with dangling cases." ACL, 2021.

---

> > ### Comment · Reviewer_sPPq · 2023-11-23
> >
> > Thanks for the thorough response. It would be nice to have these results in the draft. Since my score is already 8, I keep it unchanged.

---

> > > ### Author Response · Authors · 2023-11-23
> > >
> > > Thank you once again for your insightful suggestions. We are truly grateful for your unwavering support.

---

### Official Review · Reviewer_ALxa · 2023-10-29

**Soundness:** 2 fair
**Presentation:** 2 fair
**Contribution:** 2 fair
**Rating:** 6
**Confidence:** 3

**Summary:**

This paper studies the task of entity alignment. The authors introduce a generative EEA framework, leveraging the mutual variational autoencoder (M-VAE) to facilitate the encoding and decoding of entities between source and target KGs. A series of experiments have been executed to ascertain the efficacy of the GEEA, and the results affirm the prowess of the model.

**Strengths:**

The experiments showcased provide evidence of the efficacy of the GEEA model.

**Weaknesses:**

- The paper would benefit from enhanced clarity. Several concepts are mentioned without a clear definition, leading to potential confusion for readers. See the questions listed below for specifics.
- The objective of prior reconstruction could be made more comprehensible. There's ambiguity regarding the priors of features in different sub-embeddings. Specifically, when dealing with images, how does one retrieve its original, tangible feature?
- The paper could provide a more extensive set of experiments to offer insights into the rationale behind the design of individual components.

- Eq. (19)  is missing a right parenthesis.

**Questions:**

- Could you elaborate on what constitutes the multi-modal information within the knowledge graph (KG) area?
- What types of attribute information are being referred to in this context?
- How would you define "sub-embeddings" in the framework?
- What does $\mathcal{L}_{mf}$ represent within the paper?

---

> ### Author Response · Authors · 2023-11-13
>
> Many thanks for your helpful feedback and suggestions. We will carefully incorporate them into our paper.
>
> ### Weaknesses:
>
> - **Several concepts are mentioned without a clear definition as listed in **Questions**.**
>
>     We appreciate your constructive feedback and hope that our responses to **Questions** have adequately addressed your concerns.
>
> - **The objective of prior reconstruction: when dealing with images, how does one retrieve its original, tangible feature?**
>
>     Sorry for the confusion. As stated in Section 3.6 Implementation, we move the details of reconstructing the tangible features of each module to Appendix B due to space constraints. Specifically, the reconstruction of neighborhood and attribute information can be viewed as multi-label classification. As for image reconstruction, we also hightlight the process in Section 4.3 Entity Synthesis Results. We state, "Since the image features are pretrained embeddings of a vision model, we use the image corresponding to the nearest neighbor of the reconstructed feature as the output image." In the revision, we have added a breif introduction to the reconstruction of images in Section 3.6.
>
> - **The paper could provide a more extensive set of experiments to offer insights into the rationale behind the design of individual components.**
>
>     Thanks for your suggestion. We conducted ablation studies to assess the effectiveness of each component and present the results in Table 5. Specifically, we evaluated various alternative methods by removing different components while keeping all parameter settings identical to GEEA. We assessed the performance on both tasks and found that the complete GEEA achieved the best results, with any module removal leading to a performance loss.
>
>     Additionally, we have incorporated several new experiments in the revision, thanks to the valuable comments from all reviewers. These experiments contribute to a better understanding of our method. Please let us know if you have any further suggestions.
>
> - **Eq. (19) is missing a right parenthesis.**
>
>     Thank you very much. We have fixed it.
>
> ### Questions:
>
> - **Could you elaborate on what constitutes the multi-modal information within the knowledge graph (KG) area?**
>
>     Sorry for the confusion. We have clarified the description in Section 4.1 Experiment Settings in the revised version. Our paper considers three different modalities: relational graphs (including the relations and entities linked to each entity), (textual) attributes, and images. We also present the statistics of each modality in each dataset in Table 7.
>
> - **What types of attribute information are being referred to in this context?**
>
>     Similar to previous works, we treat attribute information as bags of words and represent them using pre-trained word embeddings.
>
> - **How would you define "sub-embeddings" in the framework?**
>
>     We have updated Section 2.1 Embedding-based Entity Alignment to include the definitions of "sub-embedding" and "joint embedding" in the revised version. The term "sub-embeddings" is introduced in the introduction section (the second paragraph and Figure 1). It refers to the output embeddings generated by different encoders. "The encoder module uses different encoders to encode multi-modal information into low-dimensional embeddings. The fusion layer then combines these sub-embeddings to a joint embedding as the output."
>
>
> - **What does $\mathcal{L}_{m_f}$ represent within the paper?**
>
>     The reviewer may have overlooked the explanation on the next page, specifically after Equation (22). "where $\mathcal{F} = \{x\rightarrow x, y\rightarrow y, x\rightarrow y, y\rightarrow x\}$ is the set of all flows, and $\{g, a, i, ...\}$ is the set of all available modalities."  In this context, $m$ denotes the modality (e.g, $g$), and f denotes the flow (e.g., $x\rightarrow y$). $\mathcal{L}_{m_f}$ is similar to
>
>     $\mathcal{L}_{g, x\rightarrow y}$, which is defined in Equation (19) (We have to insert a newline here because the original Latex formula collapsed).

---

> > ### Comment · Reviewer_ALxa · 2023-11-18
> > **Thank you to the authors for their detailed response**
> >
> > Thank you to the authors for their detailed response. Your revisions have addressed some of my questions and concerns, and have enhance the paper's clarity. As a result, I am considering increasing my review score. However, further improvements are necessary for the paper to reach the acceptance threshold for ICLR. Specifically, there are still numerous instances of improper writing and grammatical errors. To highlight a few:
> >
> > 1. Improper use of references. The first sentence in the Introduction cites over 15 papers, which seems excessive and unnecessary.
> > 2. Some grammar issues:
> > - The phrase "are friendly to development and deployment" should be revised to "are friendly for development and deployment."
> > - "low-dimension vectors" should be corrected to "low-dimensional vectors."
> > - Clarification is needed regarding which models are referred to in some "existing EEA models." or "existing EEA works"
> > - "one major issue with the existing GAN-based method is mode collapse" so this issue is only for one model proposed by (Srivastava et al.,2017)? How about other models? Should it be the "existing GAN-based methods"?
> >
> > Furthermore, the definitions of modalities (i.e., relational graph information, attribute information, and image information) are presented in an unclear manner. It is not immediately apparent how images can be represented as discrete features.
> >
> > The paper could also benefit from a stronger motivation, particularly regarding the necessity of a generative approach to the EA task. Additionally, the rationale behind the Entity Synthesis task should be more thoroughly explained in the introduction. How it related to the EA task?
> >
> > Lastly, the structure of the paper requires refinement. Essential preliminary knowledge is introduced after Section 2, which already employs the VAE for modeling. This sequence may lead to confusion for readers unfamiliar with the background concepts.

---

> > > ### Author Response · Authors · 2023-11-19
> > >
> > > Thank you very much for your valuable and detailed response. We truly appreciate the opportunity to address your concerns and provide further clarifications.
> > >
> > > **Writing and grammartical errors.**
> > >
> > > We have carefully addressed the identified typos and conducted a thorough review of the entire paper to ensure the clarity.
> > >
> > > 1. Improper use of references.
> > >
> > >     We have revised the sentence by reducing the number of cited papers.
> > >
> > > 2. Grammar issues.
> > >
> > >     We have carefully corrected all typos and highlighted them in blue in the new revised version. Thank you again for your detailed comments.
> > >
> > >
> > > **The definitions of modalities are presented in an unclear manner. It is not immediately apparent how images can be represented as discrete features.**
> > >
> > > We have revised Section 2.1 as Preliminaries, which now consists of three subsubsections: Entity Alignment, Entity Synthesis, and Generative Models. In the Entity Alignment section, we have added an example to illustrate multi-modal settings:
> > >
> > > For instance, the relational graph feature of an entity *Star Wars (film)* is represented as triplets, such as *(Star Wars (film), founded by, George Lucas)*. Similarly, the attribute feature is represented as attribute triplets, e.g., *(Star Wars (film), title, Star Wars (English))*. For the image feature, we follow the existing multi-modal EEA works to use a constant pretrained embedding from a vision model as the image feature of *Star Wars (film)*.
> > >
> > > We have also added a similar example in the Entity Synthesis section:
> > >
> > >   For instance, to reconstruct the neighborhood (or attribute) information of *Star Wars (film)* from its embedding, we can leverage a decoder module to convert the embedding into a probability distribution of all candidate entities (or attributes). As the image features are constant pretrained embeddings, we can use the image corresponding to the nearest neighbor of the reconstructed image embedding of *Star Wars (film)* as the output image.
> > >
> > > **The paper could benefit from a stronger motivation, particularly regarding the necessity of a generative approach to EA and the rationale behind Entity Synthesis.**
> > >
> > > Thank you very much for your suggestion. We have updated the introduction section to include an explanation of the motivation for studying a generative approach:
> > >
> > > In this paper, we demonstrate that adopting a generative perspective in studying EEA allows us to interpret negative sampling algorithms and GAN-based methods. Furthermore, we provide theoretical proof that optimizing the generative objectives contributes to minimizing the EEA objective, thereby enhancing overall performance.
> > >
> > > We have also included an example to illustrate the rationale behind entity synthesis:
> > >
> > > In fact, entity alignment is not the ultimate aim of many applications. The results of entity alignment are used to enrich each other's KGs, but there are often entities in the source KG that do not have aligned counterparts in the target KG, known as *dangling entities*.
> > > **For instance, a source entity *Star Wars (film)* may not have a counterpart in the target KG, which means we cannot directly enrich the target KG with the information of *Star Wars (film)* via entity alignment. However,
> > > if we can convert entities like *Star Wars (film)* from the source KG to the target KG,** it would save a major expenditure of time and effort for many knowledge engineering tasks, such as knowledge integration and fact checking.
> > >
> > >
> > > **The structure of the paper requires refinement. Essential preliminary knowledge is introduced after Section 2.**
> > >
> > > Great idea. As mentioned earlier, we have integrated all preliminary knowledge into Section 2.1 Preliminaries. We have moved the preliminary knowledge from Section 3 to the Generative Models subsubsection, in which we introduce the objective of generative models and use VAE as an exmaple for illustratation.

---

> > > > ### Author Response · Authors · 2023-11-22
> > > >
> > > > Dear Reviewer ALxa,
> > > >
> > > > Thank you once again for your insightful comments on our paper. As the discussion deadline is approaching, we would like to inquire whether our previous feedback has sufficiently addressed your concerns. If you have any additional insights or suggestions, we would greatly appreciate the opportunity to incorporate them into our revision. If most of your concerns have been addressed, we kindly request you to reconsider your evaluation.
> > > >
> > > > Thank you for your time and consideration.
> > > >
> > > >
> > > > Best regards,
> > > >
> > > > Authors

---

> > > > > ### Author Response · Authors · 2023-11-23
> > > > >
> > > > > Dear Reviewer ALxa,
> > > > >
> > > > > We would like to kindly remind you that the deadline (**November 22nd**) for discussion phase is approaching. We understand that you have a busy schedule and numerous responsibilities, and we sincerely appreciate the time and effort you have dedicated to reviewing our paper.
> > > > >
> > > > > We would like to inquire about your previous consideration for improving the rating of our paper. Your feedback is of great importance to us, and we eagerly await any insights or updates you can provide before the deadline.
> > > > >
> > > > > Thank you once again for your valuable contribution to our paper. We look forward to hearing from you soon.
> > > > >
> > > > > Best regards,
> > > > >
> > > > > Authors

---

### Official Review · Reviewer_gcSh · 2023-11-02

**Soundness:** 2 fair
**Presentation:** 2 fair
**Contribution:** 2 fair
**Rating:** 6
**Confidence:** 3

**Summary:**

The authors introduce the entity synthesis task and propose an M-VAE model that can convert entity embeddings back to the native concrete features. They also propose prior reconstruction loss and post-reconstruction loss to control the generation process. Empirical results show that entity synthesis has a positive effect on entity alignment.

**Strengths:**

1. Through theoretical analysis from the perspective of generative models, the authors point out that generative objectives contribute to the optimization of EEA models.
2. A generative EEA framework is proposed. By introducing reconstruction loss and distribution matching loss, GEEA further improves the performance of previous EEA models.

**Weaknesses:**

The author needs to briefly introduce the metrics of alignment prediction. Is it consistent with the EEA model used in GEEA?

The proposed M-VAE model is a generative model, but there are no other generative models compared to baseline models. It's better to compare with some GAN-based models like NeoEA("Understanding and improving knowledge graph embedding for entity alignment." International Conference on Machine Learning. PMLR, 2022.)

**Questions:**

GEEA is a general method, but all experiments are completed in multi-modal settings. In a single-modal scenario, will GEEA still be competitive?
The time and memory overhead of GEEA should be reported.
The number of entities in each data set is 15k. Can GEEA be run on larger datasets(such as 100k)?
Additionally, there are some other problems:
There are two "right hand" in the above line of Eq.43, one of them should be "left hand".
The weight of prediction matching loss is not included in Table 6.

---

> ### Author Response · Authors · 2023-11-13
>
> Thank you very much for your detailed and insightful comments. We have addressed your concerns below and hope our responses provide clarity:
>
> ### Weaknesses:
>
> - **The author needs to briefly introduce the metrics of alignment prediction. Is it consistent with the EEA model used in GEEA?**
>
>     Many thanks. We have incorporated an introduction to the metrics employed in entity alignment experiments, which align with those used in the EEA models. Specifically, we use Hits@1 and Hits@10 to assess the proportion of target entities that appear within the top 1 and top 10, respectively. We also employ MRR to measure the mean reciprocal ranks of the target entities.
>
>
> - **It's better to compare with some GAN-based models like NeoEA [1].**
>
>     Great idea. NeoEA is an impressive work which we have introduced in Section 2.3 and Section 5. However, it is important to note that NeoEA is not a typical generative model as it does not have the ability to generate new entities. Instead, NeoEA trains a discriminator to distinguish between entities from the source KG and others. It achieves this by sampling entities from the respective KGs and employing a Wasserstein-distance-based loss to train the discriminator and the EEA model in an adversarial manner.
>
>     To compare the performance of the proposed GEEA with NeoEA, we adapted NeoEA to MCLEA (also used in GEEA) based on the source code provided in its official repository. The results are presented in the following table:
>
>
>     | Methods  | ZH-EN Hits@1 | ZH-EN Hits@10 | ZH-EN MRR | JA-EN Hits@1 | JA-EN Hits@10 | JA-EN MRR | FR-EN Hits@1 | FR-EN Hits@10 | FR-EN MRR |
>     |---|:---:|:---:|:---:|:---:|:---:|:---:|:---:|:---:|:---:|
>     | EVA  | .680 | .910 | .762 | .673 | .908 | .757 | .683 | *.923* | .767 |
>     | MSNEA  | .601 | .830 | .684 | .535 | .775 | .617 |.543 | .801 | .630 |
>     | MCLEA  | .715 | .923 | .788 | .715 | .909 | .785 | .711 | .909 | .782|
>     | NeoEA (MCLEA)  | *.723* | *.924* | *.796* | *.721* | .909 | *.789* | *.717* | .910 | *.787* |
>     | GEEA  | **.761** | **.946** | **.827** | **.755** | **.953** | **.827** | **.776** | **.962** | **.844** |
>
>
>     Although NeoEA slightly improves the performance of MCLEA, the results are still significantly lower compared to those achieved by GEEA. We believe this is due to the fact that NeoEA was primarily designed for single-modal models, focusing on enhancing relational graph embedding only.
>
>
> ### Questions:
>
> - **In a single-modal scenario, will GEEA still be competitive?**
>
>     We first remove the image encoder from multi-modal EEA models. The results are shown in the following table:
>
>     | Methods  | ZH-EN Hits@1 | ZH-EN Hits@10 | ZH-EN MRR | JA-EN Hits@1 | JA-EN Hits@10 | JA-EN MRR | FR-EN Hits@1 | FR-EN Hits@10 | FR-EN MRR |
>     |---|:---:|:---:|:---:|:---:|:---:|:---:|:---:|:---:|:---:|
>     | EVA  | .680 | .910 | .762 | .673 | .908 | .757 | .683 | *.923* | .767 |
>     | MSNEA  | .601 | .830 | .684 | .535 | .775 | .617 |.543 | .801 | .630 |
>     | MCLEA  | *.715* | .923 | *.788* | *.715* | .909 | *.785* | .711 | .*909* | .782|
>     | MCLEA w/o image  | .658 | .915 | .726 | .662 | .904 | .740 | .662 | .902 | .747 |
>     | GEEA  | **.761** | **.946** | **.827** | **.755** | **.953** | **.827** | **.776** | .**962** | **.844** |
>     | GEEA w/o image | .709 | *.929* | .782 | .708 | *.935* | .784 | *.717* | *.946* | *.796* |
>
>
>     Notably, our GEEA without the image encoder still achieves state-of-the-art performance on several metrics, such as Hits@10.
>
>
> - **Can GEEA be run on larger datasets(such as 100k)?**
>
>     Then, the OpenEA 100K datasets [2] do not have a multi-modal version. However, it is still interesting to explore the performance of GEEA with single-modal EEA models on OpenEA 100K, similar to NeoEA. We conducted experiments following the NeoEA and present the results in the following table:
>
>     | Methods  | EN-FR 100K Hits@1 | EN-FR 100K MRR | EN-DE 100K Hits@1 | EN-DE 100K MRR | DBPedia-WikiData Hits@1 | DBPedia-WikiData MRR | DBPedia-Yago Hits@1 | DBPedia-Yago MRR |
>     |---|:---:|:---:|:---:|:---:|:---:|:---:|:---:|:---:|
>     | SEA  | .225 | .314 | .341 | .421 | .291 | .378 | .490 | .578 |
>     | NeoEA (SEA)  | *.254* | *.345* | *.364* | *.446* | *.325* | *.416* |*.569* | *.651* |
>     | GEEA (SEA) | **.269** | **.355** | **.377** | **.459** | **.349** | **.436** | **.597** | .**685** |
>
>     It is clear that our method can significantly enhance the performance of SEA, which can be attributed to the more stringent objectives analyzed in Section 2. We have also included all the aforementioned tables in Appendix C of the revision.

---

> > ### Author Response · Authors · 2023-11-13
> >
> > - **The time and memory overhead of GEEA should be reported.**
> >
> >     We present the memory cost in Table 2, where our GEEA utilizes more parameters with the same hidden size and layer number for EEA models. However, we have also developed a variant called GEEA (small), which employs fewer parameters and still outperforms existing methods significantly.
> >
> >     In Figure 3, we illustrate the MRR results w.r.t. training epochs for different methods, where the performance of GEEA reaches its peak with fewer epochs. Nevertheless, we acknowledge that our method requires more training time per epoch. Hence, we have updated Table 2 to include the total training time:
> >
> >     | Methods  | # Parameters (M) | Training time (s) | FB15K-DB15K Hits@1 | FB15K-DB15K Hits@10 | FB15K-DB15K MRR |
> >     |---|:---:|:---:|:---:|:---:|:---:|
> >     | EVA  | **10.2** | 1,467.6 | .199 | .448 | .283 |
> >     | MSNEA  | 11.5 | 775.2 | .114 | .296 | .175 |
> >     | MCLEA  | 13.2 | 285.4 | .295 | .582 | .393 |
> >     | GEEA (small) |  *11.2* | **217.3** |  *.322* | *.602* | *.417* |
> >     | GEEA | 13.9 | *252.4* | **.343** | **.661** | **.450** |
> >
> >     Our method still holds advantages in terms of training time compared to the baselines.
> >
> >
> > - **Typos: There are two "right hand" in the above line of Eq.43, one of them should be "left hand". The weight of prediction matching loss is not included in Table 6.**
> >
> >     Thank you for your thorough and insightful comments. We have addressed the identified typos and highlighted them in red in the revised version.
> >
> >
> > [1] "Understanding and improving knowledge graph embedding for entity alignment." ICML, 2022.
> >
> > [2] "A benchmarking study of embedding-based entity alignment for knowledge graphs." VLDB, 2020.

---

> ### Comment · Reviewer_gcSh · 2023-11-22
>
> Thanks for your detailed response. I would like to increase my score.

---

> > ### Author Response · Authors · 2023-11-23
> >
> > We sincerely appreciate your increased rating and recognition of the efforts we put into addressing your concerns. Your contribution to our work is highly valued and greatly appreciated.

---

### Author Response · Authors · 2023-11-13

Dear all reviewers:

We sincerely appreciate the time and effort you have dedicated to reviewing our paper.

We would like to express our gratitude to Reviewers gcSh for suggesting the inclusion of a new baseline and dataset. We have incorporated these suggestions and conducted corresponding experiments in the revised version.

We are also grateful to Reviewers ALxa for providing suggestions on improving the details of the multi-modal settings. We have updated the relevant paragraphs with additional explanatory content.

Furthermore, we extend our special thanks to Reviewer sPPq for recommending the application of our method to dangling entity detection. We have conducted experiments and we are pleased to report that some preliminary results are highly promising.

Thanks again to all reviewers. Your comments are invaluable in helping us enhance the quality of our paper, and we have uploaded the revised version with modified content marked.



Best Regards,

Authors

---

### Meta-Review · Area_Chair_UbPv · 2023-12-05

**Metareview:**

The paper introduces a generative framework for embedding-based entity alignment (EEA), leveraging a mutual variational autoencoder (M-VAE). This approach facilitates the encoding and decoding of entities between source and target knowledge graphs (KGs), and extends to the generation of new entities. The theoretical and empirical analyses demonstrate the effectiveness of this generative approach in both entity alignment and entity synthesis tasks. Reviewers collectively appreciate the paper's innovative approach and its theoretical justification. Strengths highlighted include the successful application of a generative model to EEA, introduction of a novel entity synthesis method for the dangling entity problem, and comprehensive evaluations demonstrating the method's efficacy. Despite some noted weaknesses like the need for clearer explanations and broader experimentation, there's a consensus among the reviewers on the paper's significant contribution to the field.

**Justification For Why Not Higher Score:**

The problem itself is not new.

**Justification For Why Not Lower Score:**

All reviewers have agreed to accept this paper.

---

### Decision · Program_Chairs · 2024-01-16

Accept (poster)